# Failure of DNA double-strand break repair by tau mediates Alzheimer's disease pathology in vitro

Megumi Asada-Utsugi[1,2,3], Kengo Uemura[4], Takashi Ayaki[4], Maiko T. Uemura[4,5], Sumio Minamiyama[1,3], Ryota Hikiami[1,3], Toshifumi Morimura[6], Akemi Shodai[1], Takatoshi Ueki[7], Ryosuke Takahashi [4], Ayae Kinoshita[2] & Makoto Urushitani [1,3✉]

DNA double-strand break (DSB) is the most severe form of DNA damage and accumulates with age, in which cytoskeletal proteins are polymerized to repair DSB in dividing cells. Since tau is a microtubule-associated protein, we investigate whether DSB is involved in tau pathologies in Alzheimer's disease (AD). First, immunohistochemistry reveals the frequent coexistence of DSB and phosphorylated tau in the cortex of AD patients. In vitro studies using primary mouse cortical neurons show that non-p-tau accumulates perinuclearly together with the tubulin after DSB induction with etoposide, followed by the accumulation of phosphorylated tau. Moreover, the knockdown of endogenous tau exacerbates DSB in neurons, suggesting the protective role of tau on DNA repair. Interestingly, synergistic exposure of neurons to microtubule disassembly and the DSB strikingly augments aberrant p-tau aggregation and apoptosis. These data suggest that DSB plays a pivotal role in AD-tau pathology and that the failure of DSB repair leads to tauopathy.

[1] Department of Neurology, Shiga University of Medical Science, Seta-Tsukinowa-cho Otsu, Shiga 520-2192, Japan. [2] School of Health Sciences, Graduate School of Medicine, Kyoto University, Kyoto, Japan. [3] Molecular Neuroscience Research Center, Shiga University of Medical Science, Otsu, Japan. [4] Department of Neurology, Kyoto University Graduate School of Medicine, Kyoto, Japan. [5] Center for Neurodegenerative Disease Research, Perelman School of Medicine at the University of Pennsylvania, Philadelphia, PA, USA. [6] Research Center for Animal Life Science, Shiga University of Medical Science, Otsu, Japan. [7] Department of Integrative Anatomy, Graduate School of Medical Sciences, Nagoya City University, Nagoya, Japan. ✉email: uru@belle.shiga-med.ac.jp

DNA damage is an unavoidable and relentless event for all cells, threatening the maintenance of homeostasis for cell survival. DNA repair failure is a severe phenomenon for neurons since they lack cell division[1,2]. Moreover, DNA damage accumulates in addition to the aging process, and failure of DNA repair affects transcription and translation, resulting in breakdowns for genomic integrity[1,3–7]. DNA damage occurs in response to various endogenous biological activities in cells[1,8–10], in which DNA double-strand break (DSB) is a predominant damage and can lead to cell death if not repaired[2,10,11]. In proliferating cells, the failure of DNA repair leads to cell cycle arrest and cell death[10]. In a study involving a mouse brain, DSB occurred in multiple brain regions, most abundantly in the dentate gyrus in the hippocampus, in which the repair occurred within 24 hours[9]. In another study, while focusing on a single neuron of a live zebrafish, the number of DSBs increased while the zebrafish was awake, and they decreased when they were sleeping; this phenomenon is associated with chromosome dynamics[12]. Chromosome movement in the cell are driven by cytoplasmic microtubules[13–17]. In eukaryotes, when DNA damage occurs, cytoplasmic microtubules above the nuclear membrane invade the nucleus, transform the nucleus to promote DNA mobility-dependent repair by dynein and send in DNA repair proteins. Alternatively, microtubules act as intranuclear transporter of damaged DNA to the nuclear pore complex (NPC)[16,18–21]. Thus, DNA damage repair and chromosome movement are closely related, while cytoskeleton proteins are responsible for this system. Although evidence indicates that DNA damage is implicated in the pathogenesis of neurodegenerative diseases[1,22–26], the exact role of mobility-dependent DNA repair and the microtubule in neurons is unclear.

The common pathological feature of neurodegenerative diseases is the collection of pathogenic proteins, such as amyloid β (Aβ) and microtubule-associated protein (MAPT; tau) in AD, α-synuclein in Parkinson's Disease (PD), TAR DNA-binding protein-43 (TDP-43) and fused in sarcoma (FUS) in amyotrophic lateral sclerosis (ALS). Notably, these proteins play important roles in nuclear functions. For example, α-synuclein localizes in the nucleus and is involved in the DSB repair system[27]. TDP-43 and FUS, RNA-binding proteins, contribute to the stability of mRNA and the DNA repair[28–30].

Moreover, recent reports demonstrate that DSB accumulates in the hippocampus in the brains of AD patients and AD model mice[7,31,32]. Aβ-induced oxidative stress or activation of N-methyl-D-aspartate receptors (NMDARs) induces DSB[4,9,33]. Tau protein is almost exclusively localized to the axon, however, it is also reported to localize in the nucleus to stabilize DNA[34–38]. Furthermore, there are reports which suggest that p-tau re-locates to the nuclear membrane, interacts with the NPC, and inhibits nucleus-cytosol transports in tau transgenic mice and AD brain[39], thereby resulting in cytoplasmic aggregates, which comprise of phosphorylated tau and nuclear pore proteins in neurons with neurofibrillary tangles (NFTs)[39]. In addition, autosomal dominant missense as well as splicing mutation in *MAPT* cause microtubule mediated deformation of the nucleus in inherited frontotemporal dementia (FTD)[40–42]. Several studies have reported the involvement of DSB in tau pathology. DNA-repair deficient mouse showed the p-tau pathology in the hippocampus and cotex[43]. Moreover, DNA base excision repair activity by oxidative DNA damage was enhanced in the hippocampus of 6-month-old THY-Tau22 mice, a mouse model of tauopathy[44]. However, it is unclear the mechanism for the correlation of DSB and tau pathology or NFT formation.

Since aging is the apparent risk factor for AD development, and it is also implicated in DSB, we investigated whether DNA damage is linked to tau pathology using the immunohistochemistry of AD brains and cell culture analysis. We found that DSB plays a pivotal role in AD-tau pathology and that the failure of DSB repair is linked to tauopathy.

## Results

**AD brains contain abundant DSB co-localizes with p-tau in neurons**. Several studies have reported that DSB is augmented in AD brains[4,7,31]. First, we analyzed hippocampal slices of human AD brain and non-neurodegenerative disease control brain (Table 1) to pursue DSB in the AD brain by immunohistochemistry. DSB was visualized using an anti-γH2Ax antibody, which recognizes phosphorylated histone variant H2Ax at residue Ser139, an early and essential repairment signature in DSB[45]. Consistent with previous reports[7,46], the number of DSB spots and Aβ were more extensive in the hippocampus of the AD brain than the age-matched control brains (Fig. 1a, b). To examine the regional difference of DSB vulnerability in neurons, we performed double staining for γH2Ax and MAP2 or NeuN in the temporal

**Table 1 Characteristics of human brain samples.**

| Case | Age at death(years) | Sex | Clinical diagnosis | NFT stage | CERAD |
|------|---------------------|-----|--------------------|-----------|-------|
| *Control* | | | | | |
| 1 | 74 | F | Suffocation | 0 | B |
| 2 | 68 | M | Rheumatoid arthritis | 0 | 0 |
| 3 | 70 | F | Cerebral hemorrhage | 0 | 0 |
| 4 | 68 | F | Breast cancer | 0 | 0 |
| 5 | 74 | M | Lung cancer | III | B |
| 6 | 94 | F | Disseminated intravascular coagulation | II | 0 |
| 7 | 92 | M | Prostate cancer | III | 0 |
| 8 | 86 | M | Intracranial hemorrhage | III | 0 |
| *AD* | | | | | |
| 1 | 89 | F | Alzheimer disease | V | C |
| 2 | 93 | F | Alzheimer disease | V | C |
| 3 | 85 | M | Alzheimer disease | VI | C |
| 4 | 77 | F | Alzheimer disease | V | C |
| 5 | 85 | M | Alzheimer disease | IV | C |
| 6 | 76 | F | Alzheimer disease | VI | C |
| 7 | 67 | F | Alzheimer disease | VI | C |

Clinical and histopathological information of the human brain samples used in this study. We analyzed five brains from patients with neuropathology-confirmed AD and five brains from non-neurodegenerative disease control subjects. NFT, neurofibrillary tangle; CERAD, Consortium to Establish a Registry for Alzheimer's Disease.

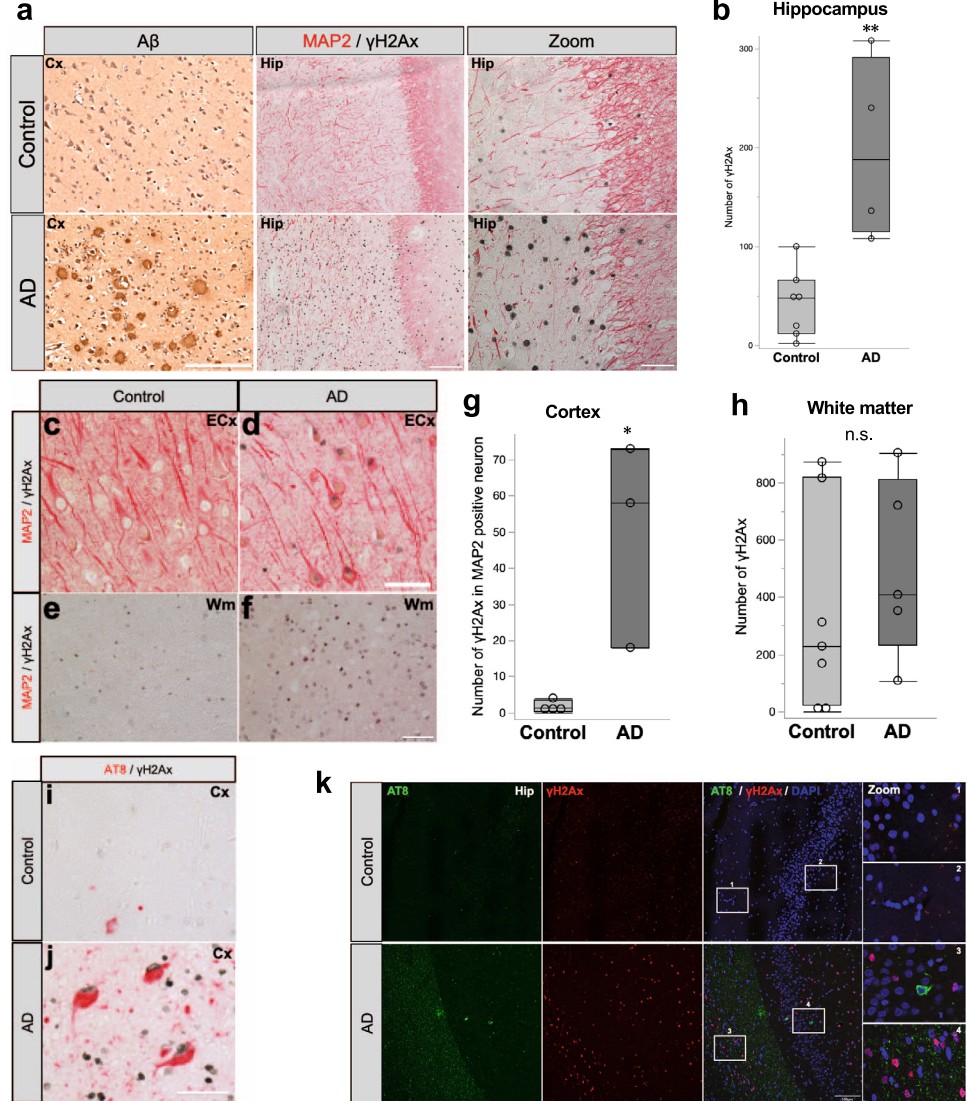

**Fig. 1 Accumulation of DSB marker in AD brains. a** Representative images of immunohistochemistry using DAB stain for Aβ in the temporal lobe cortex (Cx). Representative images of double immunohistochemistry for anti-MAP2 (neuronal marker, red) and anti-γH2Ax (phosphorylated histone variant H2Ax at residue Ser139, DSB marker, black) in the hippocampus (Hip). Scale bars = 200 μm for Aβ, 100 μm for MAP2 and γH2Ax, 50 μm for magnified images, respectively. **b** Quantitation of γH2Ax immunoreactivity in the hippocampus. Control: $n = 7$, AD: $n = 4$, n: number of sections that contained hippocampus; **$p = 0.0027$, The p-value was obtained by Student's t test. **c**, **d** Representative images of double immunohistochemistry using anti-MAP2 (red) and anti-γH2Ax (black) antibodies in the entorhinal cortex (ECx). Scale bar = 50 μm **e**, **f** Representative images of double immunohistochemistry for anti-MAP2 (red) and anti-γH2Ax (black) antibodies in the white matter (Wm). Scale bar = 50 μm. **g** Quantitation of γH2Ax immunoreactivity in MAP2 positive neuron in the cortex and entorhinal cortex. Control: $n = 7$, AD: $n = 3$, n: number of sections for γH2Ax in MAP2 positive neuron stains, *$p = 0.0177$, The p-value was obtained by Student's t test. **h** Quantitation of γH2Ax immunoreactivity in the white matter. Control: $n = 7$, AD: $n = 5$, n: number of sections for γH2Ax stains, $p = 0.4619$, The p-value was obtained by Student's t test. **i**, **j** Representative images of double immunohistochemistry for anti-AT8 (phosphorylated at S202/T205, red) and anti-γH2Ax (black) antibodies in the Cortex. Scale bar = 50 μm. **k** Representative immunofluorescent images using antibodies against p-tau (AT8, green), γH2Ax (red), and DAPI (Blue) in the hippocompus. White squares are magnifed images. Scale bar = 100 μm. DSB DNA double-strand break, AD Alzheimer's disease.

lobe cortex, entorhinal cortex, and hippocampus. In the AD brain, we observed a large number of MAP2 and NeuN-positive neurons co-localizing with γH2Ax in the entorhinal cortex and temporal lobe cortex (Fig. 1c, d, g). In contrast, most neurons were γH2Ax-negative in the hippocampus (Supplemental Fig. S1a, e). We also performed immunofluorescence analysis to identify types of cells vulnerable to DSB damage, using antibodies against NeuN, GFAP, Iba-1, ZO-1, and olig2 as markers for neurons, astrocytes microglia, endothelial cells, and oligodendrocytes, respectively. In the entorhinal cortex and temporal lobe cortex, NeuN - positive neurons co-localized with γH2Ax similar

to the result of MAP2 stains (Supplemental Fig. S1a,e). In the hippocampus, γH2Ax staining was frequently colocalized with GFAP in AD (Supplemental Fig. S1b, f). In the cerebral white matter of the temporal lobe, γH2Ax-positive cells were similarly detected in both AD and the control (Fig. 1e, f, h) and were olig2-positive (Supplemental Fig. S1d, h). Microglia and endothelial cells showed almost no co-localization with γH2Ax (Supplemental Fig. S1c, g, i). Moreover, double immunohistochemistry using AT8 (phosphorylated at S202/T205) and γH2Ax antibodies revealed that cortical cells containing phosphorylated tau (p-tau) occasionally displayed DSB in AD, where p-tau-positive cells were

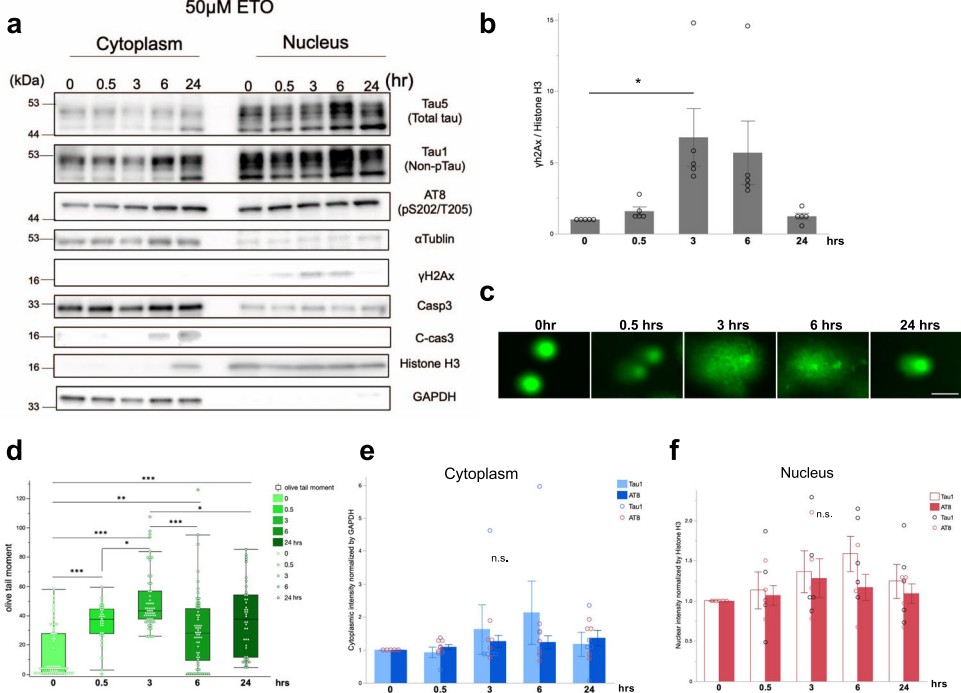

**Fig. 2 Tau increase in the nuclear fraction by DSB induction. a** Western blot analysis of tau in cytoplasmic and nuclear fractions from primary mouse cortical neuron lysates after 50 μM etoposide treatment for 0, 0.5, 3, 6, and 24 h. GAPDH and Histone H3 are used as markers for cytoplasmic and nuclear fractions, respectively. **b** Densitometric analysis of γH2Ax blots in **a**. The p-value was obtained by one-way ANOVA followed by a Tukey test ($^*P = 0.0485$, $n = 5$ independent experiments). **c** The levels of DNA damage are demonstrated by Alkaline comet images after 50 μM etoposide treatment for 0, 0.5, 3, 6, and 24 h. Scale bar = 10 μm. **d** Mean normalized olive tail moment of comets is shown in primary mouse cortical neuron treated with 50 μM etoposide treatment for 0, 0.5, 3, 6, and 24 h. The p-value was obtained by one-way ANOVA followed by a Tukey test. $^*P < 0.05$, $^{**}P < 0.01$, $^{***}P < 0.001$, 0 h $n = 67$, 0.5 h; $n = 31$, 3 h; $n = 52$, 6 h; $n = 62$, 24 h; $n = 42$. n; the number of cells in images examined by OpenComet. **e** The densitometric analysis of non-p-tau (Tau-1, white column) and p-tau (AT8, blue column) in the cytoplasm, normalized by GAPDH. $n = 5$ independent experiments. **f** The densitometric analysis of Tau1 (white column) and AT8 (red column) in the nucleus, normalized by Histone H3. $n = 5$ independent experiments.

scarce in control (Fig. 1i, j). Notably, co-localization of p-tau and DSB was quite rare in the hippocampus of AD brains (Fig. 1k).

**DSB induces tau accumulation in the nuclear fraction.** Previous reports demonstrate that the DNA moves to the nuclear pores and interacts with the inner nuclear membrane proteins for DNA repair upon severe DSB. Dissociation of damaged DNAs from the unimpaired DNA prevents aberrant recombination[21,47–49]. Hence, we examined whether subcellular localization of tau is altered in neurons when DSB occurs using etoposide, a topoisomerase II inhibitor. Primary mouse cortical neurons (DIV 7) were treated using 50 μM etoposide for 0, 0.5, 3, 6, and 24 h, and the cell lysates were separated into the cytoplasmic and nuclear fractions for Western blotting. The γH2Ax levels were highest at 3 h after treatment and decreased subsequently (Fig. 2a, b). In addition, the comet assay to quantify DSB levels showed that the damage was highest after 3 h of etoposide exposure, which is consistent with the result of WB (Fig. 2c, d). From the preliminary lethality evaluation, 24 h of exposure to 50 μM etoposide reduced cell viability by about 60 % (Supplemental Fig. S2). Active Caspase3 was detected at 6 and 24 h after treatment in cytoplasm fraction (Fig. 2a). In the cytoplasm, p-tau (AT8) gradually elevated at 6, and 24 h after treatment, wheres non-p-tau (Tau-1; dephosphorylated at S195/S198/S199/S202) levels were the highest at 6 h and declined at 24 h after DSB (Fig. 2a, e). In the nuclear fraction, p-tau and non-p-tau showed a similar expression profile, which increased at 6 h after treatment (Fig. 2a, f). This result was similar to a previous study that etoposide treatment increased total tau in the nucleus[50]. On the contrary, DSB

induction by ultraviolet (UV) exposure for 5- and 10-mins increased tau species, including total, non-phosphorylated, and phosphorylated tau in both nuclear and cytoplasmic fractions at 5 mins (Supplemental Fig. S3a). These results indicate that DSB induces nuclear accumulation of non-p-tau, which subsequently increases p-tau. UV exposure might accelerate tau accumulation due to the disproportionate effect. Since primary mouse cortical neurons cultured for 14 days also showed similar results that non-p-tau and p-tau in the nuclear fraction increased in an etoposide concentration-dependent manner (Supplemental Fig. S3b). We decided to use primary neurons of DIV 7 for the following experiments.

**Cytosolic non-p-tau accumulates around the nucleus in neurons after DSB induction.** Thereafter, we performed immunocytochemical analysis to determine the subcellular location around the nuclei, where non-p-tau had accumulated by DSB induction. After treatment with 50 μM etoposide for 6 h., the primary mouse cortical neurons (DIV 7) were analyzed by immunofluorescence staining using Tau-1 and anti-γH2Ax antibodies. γH2Ax-positive neurons were significantly increased by etoposide exposure, in which the population of γH2Ax and Tau1double-positive neurons was markedly highest (Fig. 3a–c). Intriguingly, the etoposide treatment increased Tau-1 staining in the cytosol, particularly surrounding the nucleus as shown in XZ and 3D images (Fig. 3a). Furthermore, etoposide treatment also significantly augmented Tau1 intensity in the DAPI area (Fig. 3d). These results suggest the possibility that DSB recruits tau to the nuclear periphery. We next performed immunoelectron

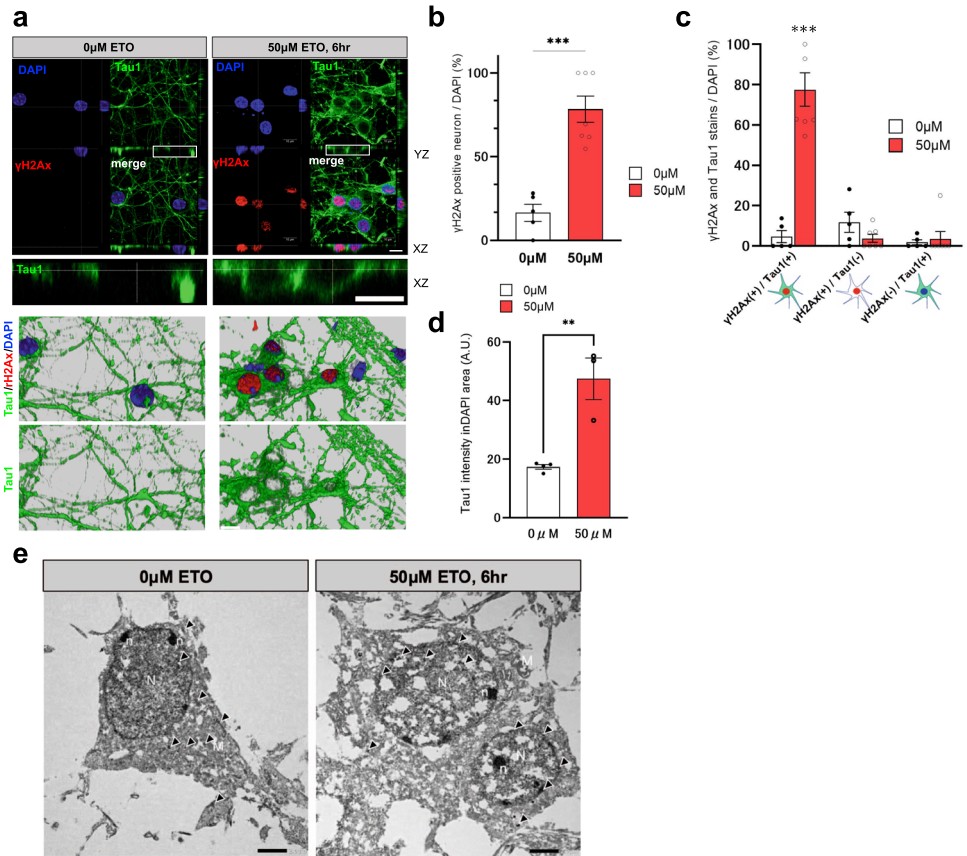

**Fig. 3 Increase in non-p-tau around the nuclear membrane of neurons by DSB induction. a** Top panels: Representative XY, YZ, and XZ projections of immunofluorescent images using antibodies against non-p-tau (Tau-1, green), γH2Ax (red), and DAPI (Blue). Mice primary neurons were treated with 50 μM etoposide or PBS (0 μM ETO) for 6 h (50 μM ETO, 6 h). White boxes show magnified XZ projections of Tau-1 staining to clarify the distribution of non-p-tau. The bottom panels show 3D reconstruction images of immunostaining. Scale bar = 10 μm. **b** The population of γH2Ax-positive neurons per total neurons labeled with DAPI in non-treatment (white column, n = 5) and 50 μM etoposide-treatment for 6 h (red column, the number of images examined = 7). n: number of images examined; **p = 0.0001, The p-value was obtained by Student's t test. **c** Quantification shows that the percentage of cells with Tau-1-γH2Ax double-positive neurons were exclusively high in 50 μM etoposide treatment (γH2Ax+/Tau-1+). Statistical significance was determined by two-way ANOVA followed by a Tukey test. n: number of images examined. ***p < 0.0001. **d** Quantification of the Tau-1 intensity in DAPI area. White column is non-treatment (n = 4), red column (n = 3) is 50 μM etoposide treatment for 6 h. n: number of images examined; **p = 0.0041, The p-value was obtained by Student's t test. **e** Immunogold electron micrographs in non-p-tau labeled with Tau-1 antibody from primary mouse cortical neurons with non-treatment (0 μM etoposide) and 50 μM etoposide treatment for 6 h Arrowheads indicate labeling the gold particles for Tau-1. N nucleus, n nucleolus, M mitochondria. Scale bar = 2 μm.

microscopy to analyze the precise localization of non-p-tau after 6 h of etoposide treatment. In non-treated neurons, Tau-1 staining showed scattered distribution, predominantly in the cytoplasm (Fig. 3e, left panel). On the contrary, at 6 h after 50 μM etoposide treatment, the gold particles for tau-1 were observed chiefly around the nuclear membrane (Fig. 3e right panel). To confirm the accumulation of non-p-tau close to the nuclear membrane by DSB induction, we performed a proximal ligation assay (PLA) probe for non-p-tau and nuclear membrane in primary mouse cortical neurons (Supplemental Fig. S4). 10 mins for UV exposure increased PLA foci in DAPI area indicating the association between Tau1 and H3K9me3, heterochromatin marker, on the nuclear membrane[51] (Supplemental Figs. S4a, b). Moreover, treatment of 50 μM etoposide for 6 h also increased PLA foci for Tau1 and LaminB, nuclear membrane protein on the nucleus (Supplemental Figs. S4c, d). These results indicate that DSB may recruit non-p-tau to the perinuclear membrane.

**DSB-induced perinuclear p-tau is neurotoxic.** Western blotting (WB) showed that p-tau might increase in the cytoplasm for 24 h after etoposide exposure (Fig. 2a, e). Therefore, we investigated

the precise subcellular location of p-tau after DSB induction using confocal and immunoelectron microscopes. XZ images of the confocal immunofluorescent analysis showed that p-tau immunoreactivity (AT8) was enhanced around the nuclei at 24 h after 50 μM etoposide exposure (Fig. 4a). Quantitative analysis also determined that AT8 immunofluorescent intensity in the DAPI area increased at 24 h after etoposide exposure (Fig. 4b). Immunoelectron microscope showed that the gold particles for AT8 scattered in the cytosol, while etoposide treatment altered the AT8 distribution closer to the nuclear membrane (Fig. 4c). Moreover, UV exposure for 10 mins also increased PLA foci for PHF-tau (AT100; pThr212/ Ser214) and heterochomatin (Supplemental Fig. S4e, f).The induction of DSB with etoposide increased the toxic oligomeric tau stained with anti-T22 antibody in a dose-dependent manner, indicating that DSB-relevant perinuclear p-tau accumulation is linked to neurodegeneration in tauopathy (Fig. 4d and Supplemental Fig. 5).

**DSB promotes the interaction between non-p-tau and tubulin.** Previous reports demonstrated that non-p-tau accelerates the polymerization of microtubules[52], whereas p-tau amplifies the

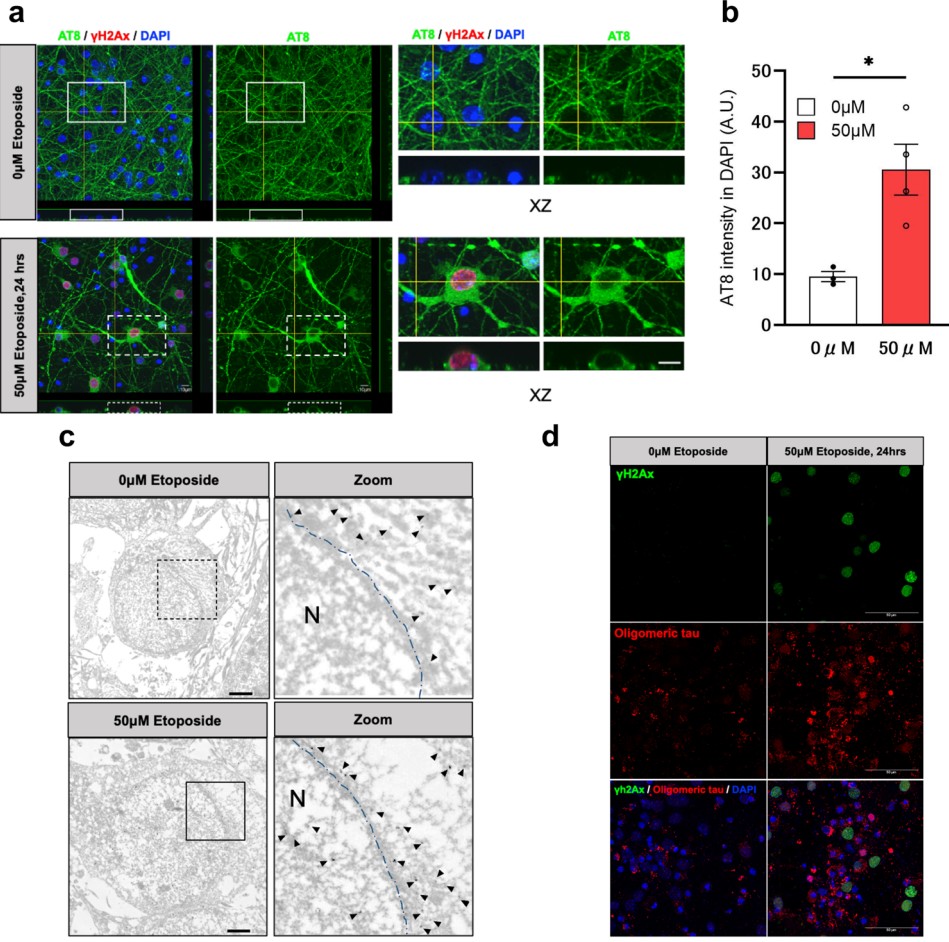

**Fig. 4 Accumulation of p-tau and toxic tau around the nuclear membrane of neurons by DSB induction. a** Left panels: representative XY, YZ, and XZ projections of immunofluorescent staining using antibodies against p-tau (AT8, green) and γH2Ax (red) and DAPI (Blue) with non-treatment (0 μM etoposide) and 50 μM etoposide treatment for 24 h (50 μM Etoposide, 24 h). Right panels show magnified XZ projections of white solid dotted squares in the left panels. Scale bar = 10 μm. **b** Quantification of the AT8 intensity in DAPI area. White column is non-treatment ($\underline{n}$ = 3), red column ($n$ = 4) is 50 μM etoposide treatment for 24 h. $n$: number of images examined; *$p$ = 0.0168, The $p$-value was obtained by Student's $t$ test. **c** Immunogold electron micrographs from primary mouse cortical neurons with non-treatment (0 μM etoposide) and 50 μM etoposide treatment for 24 h (50 μM etoposide, 24 h.), showing AT8 positive p-tau around the nuclear membrane (AT8). Zoom panels show enlarged projections of dotted square and solid square. Arrowheads indicate gold particles for AT8 antibody. N nucleus; Blue dash line, nuclear membrane. Scale bar = 2 μm. **d** Immunofluorescence using antibodies against oligomeric tau (T22, red) and γH2Ax (green) with DMSO treatment (left panel) and 50 μM etoposide treatment for 24 h (right panel). Scale bar = 30 μm.

depolymerization[53]. Since the accumulation of non-p-tau in the perinuclear region was most prominent at 6 h after DSB induction (Fig. 3), we assumed that the interaction of non-p-tau and tubulin may increase after 6 h of DSB induction. Hence, we investigated the effect of DSB on tubulin and non-p-tau assembly. Primary mouse cortical neurons (DIV 7) were exposed to 50 μM etoposide for 0, 0.5, 3, 6, and 24 h., and thereafter, they were analyzed by PLA probe assay. Results show that the number of PLA signals (red foci) increased the most at 6 h after the etoposide treatment (Supplemental Fig. S6a and Fig. 5a, b). Moreover, co-PLA foci were exclusively adjacent to DAPI signals, and they were most noticeable at 6 h (Fig. 5c). PLA signals for p-tau and tubulin were almost undetectable (Supplemental Fig. S6b). These results indicate that DSB promotes the interaction between tubulin and non-p-tau around nuclei, enhancing the microtubule assembly, and resulting in the accumulation of non-p-tau around the nucleus.

**Tau knockdown by shRNA increases γH2A in the early stage of DSB induction**. Henceforth, to investigate the effect of tau on

DSB and repair, we performed knockdown for endogenous mouse tau by lentivirus vector-mediated shRNA for primary mouse cortical neurons. Endogenous mouse tau expression levels effectively reduced about ~60% of tau protein compared to control shRNA (Fig. 6a–c). We checked the intensity of γH2Ax on WB by comparing the control and mouse tau shRNA. At the instance when shRNA-treated primary mouse cortical neurons were exposed to etoposide for 30 min, tau-knockdown increased γH2Ax 2-fold higher than control shRNA without altering the level of active caspase-3 (Fig. 6d, e). We studied the more prolonged effect of 30 min DSB induction on the delayed cell survival, in which neurons after replacement with normal medium were cultured for subsequent 24 h. Fascinatingly, downregulation of tau reduced γH2Ax by ~60% compared to the control shRNA at 24 h (Supplemental Fig. S7a, b). Similar results were obtained for other shRNA clones (Supplemental Fig. S7c). These results indicate that tau may contribute to DSB repair by accompanying tubulin and the recruitment to the perinucleus in the early phase.

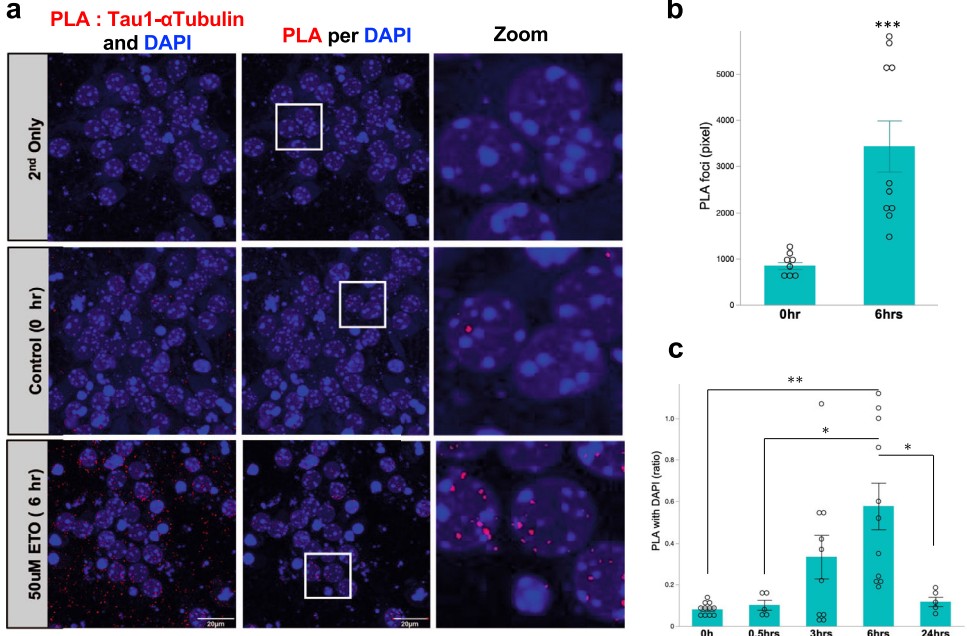

**Fig. 5 Promoted interaction between non-p-tau and α-tubulin by DSB induction. a** Left panels; PLA assay for primary mouse cortical neurons exposed to 50 μM etoposide for 0, 6 h Top panels show PLA without primary antibodies (2nd only: negative control). Nuclei were stained with DAPI. Middle panels show the extraction images of PLA foci, which are colocalized with DAPI (PLA per DAPI). The zoomed images were shown in white solid squares on the right. **b** The number of PLA foci in primary mouse cortical neurons exposed to 50 μM etoposide treatment for 0 h ($n = 9$) and 6 h ($n = 10$). The p-value was obtained by Student's t test. ***$p = 0.0008$. **c** PLA foci merged on DAPI in primary mouse cortical neurons exposed to 50 μM etoposide treatment for 0 ($n = 11$), 0.5 ($n = 5$), 3 ($n = 10$), 6 ($n = 11$) and 24 h ($n = 5$). The p-value was determined by two-way ANOVA, followed by a Tukey test. 0 h. vs 6 h, **$p = 0.0005$; 0.5 h vs 6 h, *$p = 0.011$; 24 h vs 6 h, *$p = 0.015$. n number of images examined.

**DSB and microtubule disassembly synergistically induced aberrant p-tau aggregates and caspase activation.** DSB induction enhanced interaction between non-p-tau and tubulin, implying that the enhancement of tubulin polymerization by tau may be involved in DSB repair. Since the tau phosphorylation is tightly linked to microtubule disassembly, we investigated the effect of the inhibition of microtubule polymerization under DSB conditions by examining the distribution of p-tau, using microtubule polymerization inhibitor, 2(-2,6Diisopropylphenyl)-5-hydroxy-1H-isoindole-1,3-dione (5HPP-33). After pretreatment with 2.5 μM 5HPP-33 for 30 mins, primary mouse cortical neurons were exposed to 50 μM etoposide for 24 h. It was confirmed by PLA probe assay using α-tubulin and β-tubulin antibody verified microtubule polymerization and disassembly between α-tubulin and β-tubulin by 5HPP-33, but not by DMSO treatment (Supplemental Fig. S8a, b). Interestingly, the exposure to both etoposide and 5HPP-33 induced an excessive and aberrant cytosolic accumulation of p-tau throughout the neurons, mimicking an NFT (Fig. 7a). Treatment with bleomycin, which induces DSB by a different mechanism of action, yielded the same results (Supplemental Fig. S9). Moreover, WB analysis showed that the combination of etoposide and 5HPP-33 enhanced the levels of γH2Ax and cleaved-caspase3 in detergent-soluble fraction (Fig. 7b–d), in which total tau and p-tau species was increased in the sodium dodecyl sulfate (SDS)-insoluble fraction (Fig. 7b, e). Based on these results, inhibition of microtubule polymerization under DSB might lead to the formation of NFTs and neurotoxicity.

## Discussion

We have demonstrated evidence that corroborates the involvement of DSB in tauopathy in AD using human brain and in vitro experiments. Although DSBs were observed in neurons in the postmortem AD brain, glial cells, especially oligodendrocytes, surprisingly showed more prominent DSB signals (Fig. 1e). This is likely due to the fact that neuronal DSB might be underestimated on account of increased neuronal death in the AD brain. On the other hand, most neurons cannot divide, but glial cells can[54]. Therefore, glial cells are capable of precise repair by homologous recombination when DSB occurs. In addition, it is reported that DSB repair in neuron occurs faster than in mouse embryonic fibroblasts (MEFs), possibly due to the higher DNA-dependent protein kinase activity (DNA-PK), which is the repair protein in the NHEJ repair system[55]. Neurons may have more specialized repair systems for DSB repair than glial cells.

In dividing cells, the repair of DNA damage is tightly associated with the cell cycle, in which the DNA damage response activates cell cycle checkpoints to prevent replicating mis-genetic information[56]. Contrastingly, it has been reported that non-dividing postmitotic neurons in neurodegenerative diseases, such as AD, undergo a re-entry into the cell cycle occurred for DNA repair[57–59], which is accompanied by the expression of cell cycle-related proteins[60–62].

Moreover, data from in situ PLA probe and electron microscope revealed that DSB induction recruited non-p-tau to the nuclear membrane together with the augmentation of tau-tubulin interaction (Figs. 2, 3 and 5). These findings indicate that cytoskeletal rearrangement might occur after DSB induction, which drives tau in microtubule assembly. Indeed, several reports demonstrate that p-tau is present on a mitotic spindle, which contains microtubules, during cell division of the spermatogenesis in mouse testes[63,64]. Consequently, the increase of non-p-tau and p-tau around the nuclear envelope by DSB induction might relate to microtubule reassembly for the neuronal cell cycle re-entry. Giorgio U. et al. also examined the phosphorylation state of nuclear tau in detail in proliferating pluripotent neuronal C17.2 and neuroblastoma SY5Y cells and showed an increase in pT181,

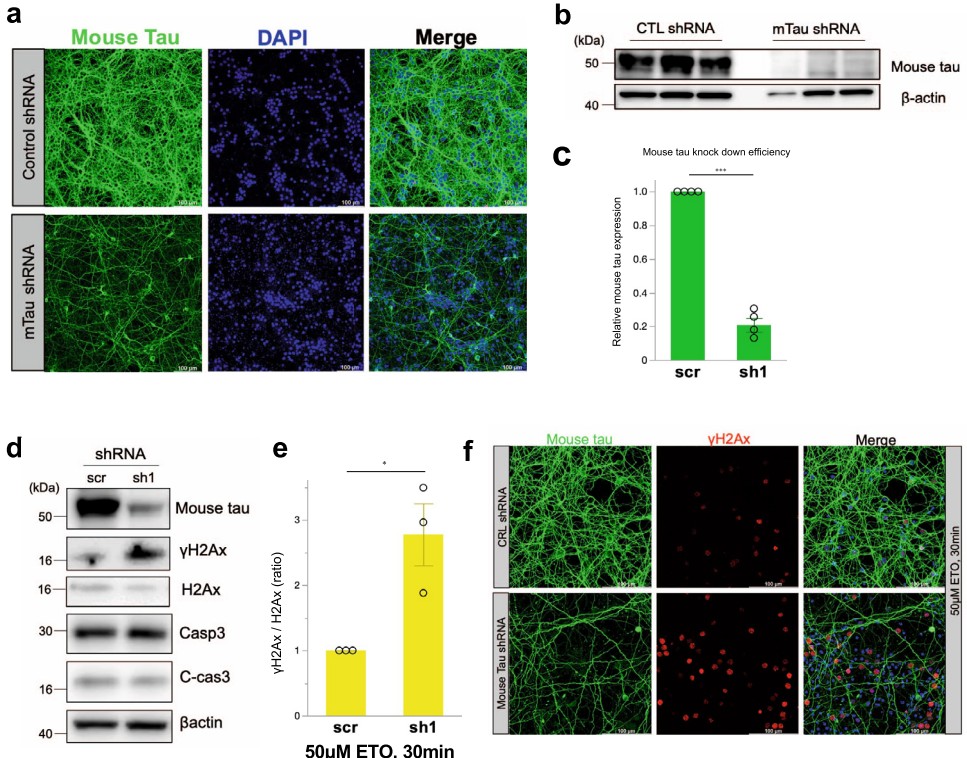

**Fig. 6 Mouse tau shRNA lentivirus knockdown exacerbates DSB. a** Confocal microscopy images of mouse tau (green) and DAPI (blue) on primary mouse cortical neurons. Primary mouse cortical neuron cultures (DIV 0) were transduced by mouse tau and control shRNA lentivirus particles, cultured for 7 days. Scale bar = 100 μm. **b** Endogenous mouse tau expression with anti-mouse tau antibody for control (CTL) and mouse tau (mTau) shRNA knockdown (n = 3). **c** Quantitation of mouse tau expression, scr: control shRNA, sh1: mouse tau shRNA colne 1, n = 4, n: number of samples for WB, ***p < 0.0001. Student's t test. **d** Western blots of control and mouse tau shRNA knockdown with 50 μM etoposide treatment for 30 min. **e** The ratio of γH2Ax/ H2Ax with 50 μM etoposide treatment for 30 min was compared between the control (scr) and mouse tau shRNA (sh1), n = 3, n: number of samples for WB, *p = 0.0367. Student's t test. **f** Confocal microscopy images of mouse tau (green), γH2Ax(red), and DAPI (blue) on primary mouse cortical neuron when endogenous mouse tau knocked down and treated with 50 μM etoposide for 30 mins. Scale bar = 100 μm.

pT212, and S404[50]. DSB by etoposide decreased p-tau and increased non-p-tau, resulting in tau translocation into the nucleus. Conversely, we showed that DSB induction increases nuclear p-tau (AT8). The apparent discrepancy might be derived from the phosphorylation sites since we used the AT8 antibody. Otherwise, our results that DSB induction increases nuclear non-p-tau agree with theirs. We presented that DSB sequentially promoted the accumulation of non-p-tau around the nuclear membrane, followed by the accumulation of p-tau (Figs. 2, 3 and 4). Oligomeric toxic tau species are also increased (Fig. 4c). Interestingly, cyclin-dependent kinase 5 (CDK5), one of the kinases responsible for tau phosphorylation, is activated by DNA damage and phosphorylates ATM as the first step in the repair signal[65]. It has been repoted that Tau is also phosphorylated by Chk1 and 2, DNA damage-activated Checkpoint kinase[66]. DSB may initially increase non-p-tau for DNA repair, but later result in the formation of cytotoxic tau species such as oligomeric tau or NFT through the activation of tau-phosphorylatd enzymes such as CDK5, Chk1, Chk2, GSK3β (glycogen synthase kinase 3β) or p38 MAPK (mitogen-activated protein kinase). Thus, further investigation regarding the role of tau-phoshorylated enzymes in DSB induction in tauopathy will contribute to understanding AD.

Based on the results of the tau knockdown study, it is strongly suggested that tau may have a role in promoting DSB repair with tubulin polymerization in the early stages of DSB development (Fig. 6). DSB was exacerbated in cultured neurons in an etoposide-free nutrient medium for 24 h after 30 mins etoposide treatment. However, the knockdown of endogenous tau rescues the neuron from DSB-related toxicity (Supplemental Fig. S4).

Interestingly, pharmacological depolymerization of tubulin and DSB induction led to accumulation of p-tau in the neuron, especially in the insoluble fraction (Fig. 7), indicating that failure in the cytoskeleton rearrangement under DSB condition might cause neurodegeneration. Considering the possibility that aging is a risk factor for sustained DSB, tau, by phosphorylation and microtubule disassembly, may exert a negative impact on DSB repair and neuronal survival.

In summary, neurons are physiologically exposed to various crises, such as aging, oxidative stress, genetic mutation, and excitotoxicity, all of which cause DNA damage. Under normal conditions, when DNA damage occurs, cytoskeletal proteins, such as microtubules, polymerize to execute DNA repair, and they might form a link with the nuclear membrane for the repair. Tau may well be involved in the rearrangement of microtubules to assist this process. However, the excessive DNA damage may cause accumulation of p-tau in the soma and may disassemble microtubules, which may exacerbate DNA damage and lead to neuronal cell death (Fig. 8).

Although we did not present the molecular basis of non-p-tau in the DNA repair, our solid evidence using human and in vitro studies indicate the tight relationship between DNA damage and tau cytopathology.

## Methods

**Human brain samples.** Human hippocampal paraffin sections were collected from the Department of Neurology at the Kyoto University Hospital. Details of the patients used in this study are summarized in Table 1. This collection was approved

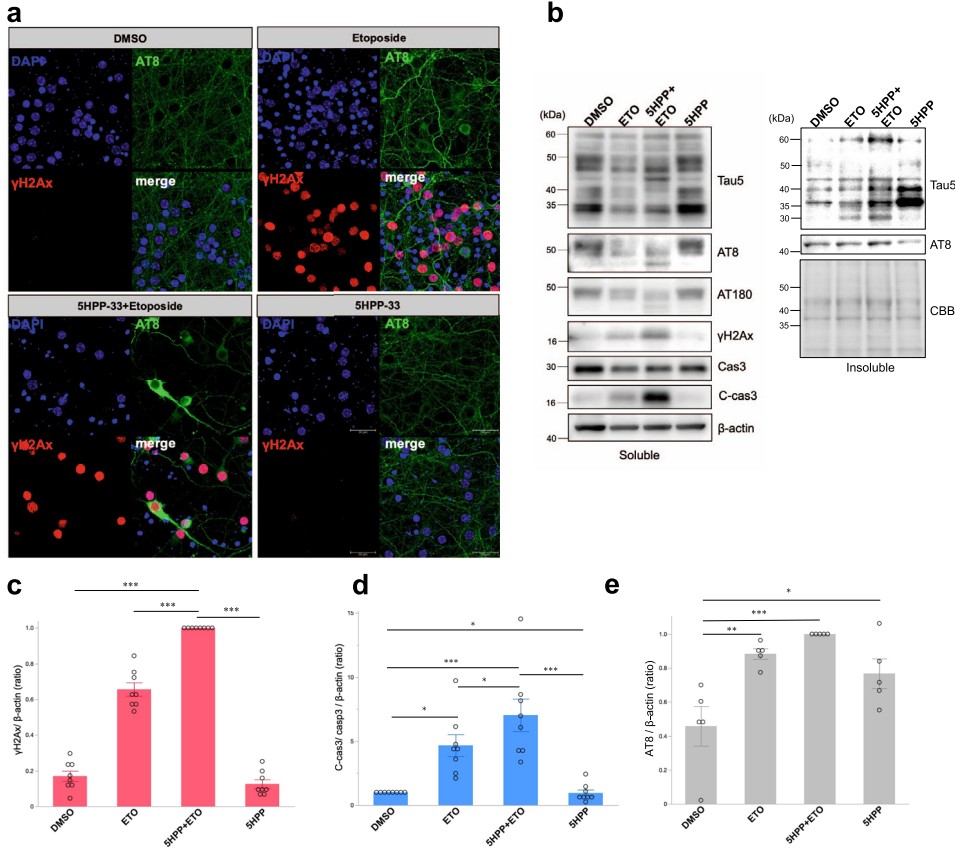

**Fig. 7 The effect of microtubule polymerization inhibitor with DSB induction. a** Primary mouse cortical neuron cultures (DIV 7) were incubated with DMSO, 50 μM etoposide (Etoposide), and 2.5 μM 5HPP-33 (5HPP-33: microtubule polymerization inhibitor) for 24 h. Neurons labeled as (5HPP-33+Etoposide) are pretreated by 2.5 μM 5HPP-33 for 30 mins, followed by incubation with 2.5 μM 5HPP-33 and 50 μM etoposide for 24 h. Immunofluorescence was obtained using antibodies against p-tau (AT8: green) and γH2Ax (red). DAPI (blue) was used for labeling the nucleus. Scale bar = 30 μm. **b** Western blot analysis of supernatant (soluble fractions, left panel) and pellet (insoluble fractions; right panel) obtained from neurons treated with DMSO, ETO (50 μM etoposide), 5HPP + ETO (50 μM etoposide treatment after pretreated by 2.5 μM 5HPP-33 for 30 mins), and 5HPP (2.5 μM 5HPP-33). Membranes were probed with anti-total tau (Tau-5), p-tau (AT8 and AT180), DSB maker (γH2Ax), caspase3 (Cas3), cleaved caspase3 (C-cas3), and β-actin antibodies. Total proteins of insoluble faction samples were stained by Coomassie brilliant blue (CBB). **c** Quantitation of γH2Ax / β-actin in soluble fraction. $n = 7$, ***$p < 0.0001$. **d** Quantitation of AT8 in insoluble fraction. $n = 4$, *$p = 0.0425$, **$p = 0.0047$, ***$p = 0.0005$. **e** Quantitation of cleaved caspase3 in soluble fraction. $n = 3$ independent experiments. DMSO vs ETO, *$p = 0.0176$; ETO vs 5HPP + ETO, *$p = 0.0496$; ETO vs 5HPP, *$p = 0.0189$; DMSO vs 5HPP + ETO, ***$p = 0.0004$; 5HPP + ETO vs 5HPP, ***$p = 0.0005$. Statistical significance was determined by two-way ANOVA, followed by a Tukey test.

by the institutional research committee, Kyoto University Graduate School, and the Faculty of Medicine, Ethics Committee.

### Immunohistochemistry and immunocytochemistry of human hippocampal sections.
Human hippocampal paraffin slices were deparaffinized in two changes of xylenes and rehydrated in graded ethanol solutions. Antigen retrieval was performed by autoclaving at 120 °C for 15 mins using Histo VT One (Nacalai Tesque, Japan). Slices were washed thrice with phosphate buffered saline (PBS) buffer (Dulbecco's PBS) with 0.05% Tween20 (PBS-T) for 5 mins. Endogenous peroxidase activity was inactivated with 0.3% $H_2O_2$ for 15 mins, washed in PBS-T buffer, and incubated with primary antibody overnight at 4 °C. After washing with PBS-T buffer, the slices were incubated with secondary antibody (Nichirei Bioscience Inc., Japan) for 30 mins at 37 °C. Slices were washed with PBS thrice; immunoreactivity was visualized with peroxidase stain DAB kit ((3,3'-Diaminobenzidine, Nacalai Tesque) and metal enhancer for DAB Stain (Nacalai Tesque). Thereafter, after washing in distilled water (DW), for the second staining, slices were incubated with another primary antibody overnight at 4 °C. After washing with PBS-T buffer, the slices were incubated with alkaline phosphatase (AP)-fused secondary antibody (Nichirei Bioscience Inc.) and were visualized with Fast Red II Substrate kit (Nichirei Bioscience Inc.). The light-microscopic analysis was performed using a Nikon micro photo-FXA microscope. For the immunofluorescence, after antigen retrieval blocking using Blocking One Histo (Nacalai Tesque, Japan), the samples were incubated with the primary antibody incubation overnight at 4 °C, subsequently with secondary antibody, CF™488A and 594 (Biotium). Images were analyzed by Leica TCS SP8 confocal microscope and the bundled software.

### Antibodies.
We used commercially available antibodies for the following antigens: Aβ (1:500, clone 6E10, BioLegend), γH2Ax (1:1000, #2577), H2Ax (1:1000, #7631), Histone H3 (1:1000, #9717), Caspase3 (1:1000, #9662), cleaved Caspase-3 (1:1000, #9661), α-Tubulin (1:1000, #2144, Cell Signaling Technology, MA), NeuN (1:500, EPR12763, abcam, UK), MAP2 (1:1000, clone Ap20, BD Biosciences, NJ), AT8 (1:1000, MN1020), AT180 (1:1000, MN1040), AT100 (1:1000, MN1060), ZO-1 (1:500, 40-2200), Tau5 (1:1000, AHB0042, Thermo Fisher Scientific, MA), γH2Ax (1:1000, host mouse, clone JBW301,#05-636), Tau-1 (1:1000, clone PC1C6, MAB3420), Olig2 (1:500, AB9610), β-actin (1:10000, A5441), H3K9me3 (1:1000, 05-499), Tau oligomeric (T22, 1:1000,#ABN454, millipore, CA), LaminB (1:1000, M-20, sc-6217), β Tubulin (1:1000, sc-5274), GAPDH (1:5000, FL-335, sc-25778, santa cruz biotech.), GFAP (1:500, G 3893, Merck, DE), mouse tau (1:1000, 012-26963), and Iba1 (1:500, 013-27691, FUJIFILM Wako Chemical Coporation, JP); mouse, rabbit and goat IgG HRP-conjugated (Jackson ImmunoResearch, PA).

### Primary mouse cortical neuron culture.
Primary neurons from wild-type (WT) mice (ICR) brains were obtained from the cerebral cortices of fetal mice (14–16 days of gestation). The cerebral cortices were washed thrice using the Hanks' Balanced Salt Solution (HBSS; Nacalai Tesque, Japan) and incubated in Accumax (Nacalai Tesque) and DNase I (Sigma-Aldrich, MO) for 5 mins at 37 °C, followed by the dissociation by gentle pipetting for the flowthrough into the 100 μm cell strainer (Greiner). Obtained cells were suspended in DMEM (Nacalai Tesque) containing 10% fetal bovine horse serum and 10% horse serum (Gibco, MA) and plated on dishes coated with 0.1% poly-ethyleneimine (Sigma) in 0.05 M boric acid buffer. After 1 h, cells were resuspended in the neurobasal medium (Gibco) containing GlutaMAX™-I(Gibco), B-27 supplement (Gibco), and 1%

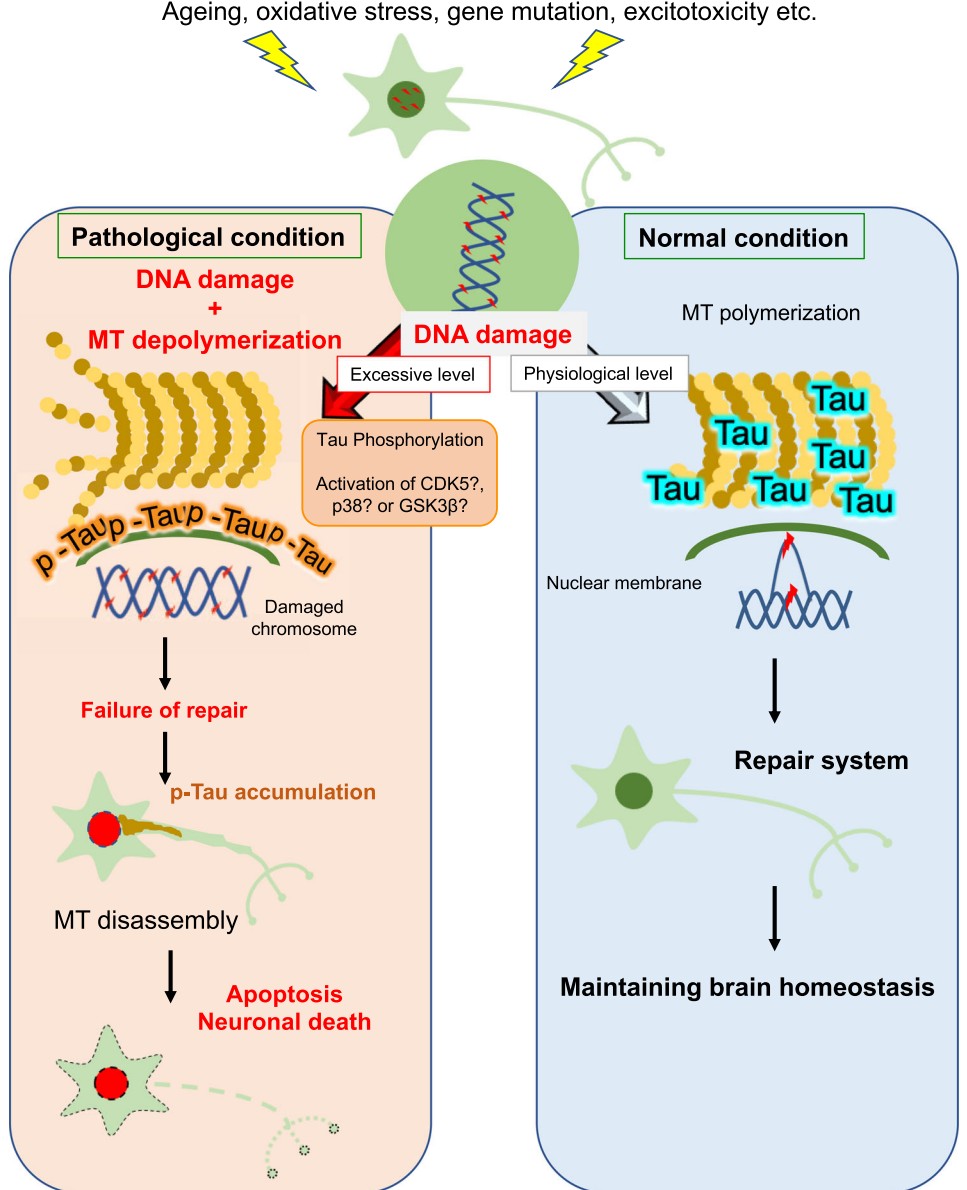

**Fig. 8 Hypothetical model of neuronal death and DNA damage repair from the perspective of the relationship between tau and microtubule polymerization.** Upon DNA damage, non-p-tau (blue) binds to microtubules at the physiological level, which promotes microtubule (MT) polymerization and induces accumulation around the perinuclear membrane to support DNA damage repair (Right: normal condition). When DNA damage reaches an excessive level, tau phosphorylation is enhanced and microtubule depolymerization occurs. Accumulation of p-tau (orange) around the perinuclear membrane suppress the movement of damaged chromosomes or inhibits the recruitment of DNA damage repair factors into the nucleus by inhibiting nuclear cytoplasmic transport. Subsequently, insoluble p-tau accumulates in the cytoplasm and axons, leading to further microtubule polymerization and neuron death (left: pathological condition).

penicillin/streptomycin (Nacalai Tesque). After two days, the medium of the culture was replaced by 10 μM Cytarabine (Ara C, Tokyo chemical industry, Japan) in a neurobasal medium and cultured for an additional seven days.

**In situ PLA**. Primary mouse cortical neurons grown for seven days in 8-well chamber slides were fixed with 4% paraformaldehyde, then permeabilized with 0.2% Triton in PBS, followed by incubation with a primary antibody for α-tubulin (rabbit) and tau-1(mouse). For the assessment of tau-α-tubulin interaction, we used a PLA, an antibody-based detection method for protein-protein interaction (Sigma-Aldrich). The nuclei were counterstained with DAPI, and the PLA signals were visualized in a confocal microscope (Olympus FV10). Incolocalization analysis, we used FLUOVIEW software (Olympus).

**Reagents for DSB induction and a microtubule polymerization inhibitor**. Primary mouse cortical neurons were exposed to etoposide (Wako Inc., Japan) and bleomycin for DSB induction after cultured for seven days. In order to inhibit

microtubule polymerization, primary mouse cortical neurons were pretreated with 2.5 μM 5HPP-33 (Wako Inc.) for 30 mins after cultured for seven days, followed by incubation with etoposide.

**SDS-PAGE and western blotting**. Protein concentrations of primary neuron lysates were determined by BCA Protein Assay Kit (Pierce Inc., Rockford, IL, USA) against albumin standards. Protein samples were prepared in 2x loading buffer (0.5 M Tris (pH 6.8), 10% SDS, 12% 2-mercaptoethanol, and 0.02% bromophenol blue) was separated to SDS-PAGE electrophoresis on a 5%–20% precast-gels (Wako Inc., Japan) and then transferred onto polyvinylidene fluoride membranes. The membranes were blocked in 5% milk in Tris Buffered Saline with Tween 20 (TBST) buffer (0.2 M Tris (pH 7.5), 1.37 M NaCl, 1% Tween) for one hour and then incubated with primary antibodies (1/1000) overnight, followed by the incubation in secondary antibodies, anti-rabbit or anti-mouse IgG Horseradish peroxidase (HRP) conjugated (Biotium) for one hour at room temperature. The membranes were washed with Tris Buffered Saline (TBS) and reacted with Chemi-Lumi One L or supper (Nacalai Tesque, Japan) for the visualization. According to

the manufacturer's instructions, we used NE-PER Nuclear and Cytoplasmic Extraction Reagents (Thermo) in nuclear and cytoplasmic fractionation.

**Electron microscopy**. The cultures were fixed with 4% paraformaldehyde in phosphate buffer saline (PBS) for 8 h., washed thrice with PBS, and permeabilized with 0.1% Triton-X100 and 1% bovine serum albumin in PBS for 10 mins. After rinsing thrice with PBS, the cultures were incubated in 1% bovine serum albumin in PBS for 30 mins and in primary antibody solution (anti-Tau-1 ab, 1:2000, 1% bovine serum albumin in PBS) for 8 h After rinsing thrice with PBS again, the cultures were incubated in secondary antibody solution (1:100, Nanogold, Nanoprobes Inc, NY, USA, with 1% bovine serum albumin in PBS) for 8 h., rinsed with PBS and enhanced with silver acetate solution for 12 mins[67]. Following this, the cultures were rinsed thrice with DW, immersed in 0.05% sodium acetate for 1 min, rinsed thrice with DW again, immersed in 0.05% gold chloride solution for 2 mins, and rinsed with DW once more. The cultures were postfixed with 0.1% osmium tetroxide in PBS for 30 mins and rinsed with DW three times. After dehydration of the cultures with graded ethanol and embedding in Epon 812 (TAAB Laboratories, UK), the culture dish was removed, and ultrathin sections (70 nm) were cut parallelly to the cell layers using an ultra-microtome (Reichert-Jung). Sections on 150 square-mesh grids covered with formvar were stained with uranyl acetate, followed by lead citrate, and examined under an electron microscope (JEOL, Japan) at 80 kV.

**shRNA knockdown of mouse tau**. We used MISSION Lentiviral Transduction Particles to knockdown mouse tau according to the target sequences defined in Sigma-Aldrich (SHCLNV, NM_010838, Clone ID TRCN0000091300) and MISSION shRNA non-target shRNA control transduction particles (SHC016V-1EA). Lentiviral particles were transduced 10 μl of $1.4 \times 10^7$ VP/mL on day 0 of primary mouse cortical neuron, which was cultured for seven days.

**Comet assay**. Primary neurons treated with etoposide were performed according to Trivigen's instructions (Trevigen CometAssay, #4250-050-K). Cell susupensions ($1 \times 10^5$ cells / ml) in PBS mixed at at a1:10 ratio with Comet LMA agarose (Trevigen) were pitetted onto CometSlides$^{TM}$ (Trevigen) and placed at 4 °C in the dark in the dark. Slides were immersed overnight in Lysis Solution (Trivegen) at 4 °C in the dark and then in Alkaline Unwinding solution (200 mM NaOH, 1 mM EDTA, pH13) for 1 hr at 4 °C. Slides were electrophoresed in Alkaline Electrophoresis solution (300 mM NaOH, 1 mM EDTA, pH13) for 40 mins at 1 V/ cm, 300 mA, at 4 °C. Slides were incubated in 70% EtOH for 5 mins, dry at 37 °C and were stained with 1 × SYBR Gold (invitrogen) at room temperature for 30 mins in the dark. Comets were visualized on BIOREVO BZ-9000 (KEYENCE) and scored using OpenComet (imageJ).

**Statistics and reproducibility**. For most analyses, data is shown as the mean ± standard error (SEM) from 2 to 4 independent experiments. Results were analyzed using one-way or two-way ANOVA, Tukey-Kramer HSD test, and Student's $t$ test. The statistical analyses were performed using JMP® 16 (SAS Institute Inc., Cary, NC, USA). The level of significance was set at $P < 0.05$ ($^*P < 0.05$, $^{**}P < 0.01$, $^{***}P < 0.001$, and ns $= P \geq 0.05$). The making graph were used by GraphPad Prism (GraphPad software 9.0 (https://www.graphpad.com/scientific-software/prism) and JMP® 16 (SAS Institute Inc., Cary, NC, USA).

**Ethics approval and consent to participate**. The human samples were taken for the project 'Pathological and biochemical studies of neurodegenerative diseases using human autopsy brain and spinal cord' (No. R1038) by the Kyoto University Ethics Committee. Informed and written consents were obtained from all individuals or their guardians, before the autopsy analysis according to the Declaration of Helsinki. All animal experiments were approved by the Animal Care and Use Committee of Shiga University of Medical Science (2020-5-8) and Kyoto University of Medical Science (Med Kyo 20017).

**Reporting summary**. Further information on research design is available in the Nature Research Reporting Summary linked to this article.

## Data availability
All data in this study is available at Mendeley Data with https://doi.org/10.17632/cy5y2wncbz.1. Uncut Western blot images are provided in Supplementary Figures. All relevant data including the numerical and statistical source data that underlie the graphs in figures are provided as Supplementary Data 1(as an Excel file). Imformation of human brain samples is provided in Suppl. Tabel 1.

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

## Acknowledgements

We thank Prof. S. Takeda (Kyoto University) for DSB induction method, K. Asamoto (Kyoto University) for immunohistochemistry and K. Tanigawa (Kyoto University) for creating graphs. We thank the Central Research Laboratory, Shiga University of Medical Science (CRL), and the Medical Research Support Center, Kyoto University Graduate School of Medicine, for technical support. This work was supported by the intramural research grant of Shiga University of Medical Science and by a research fund from Kim's Korean Ginseng CO. LTD.

## Author contributions

M.A.-U. and M.U. gratefully contributed to the study design. M.A.-U. performed experiments, analyzed data, and wrote a first draft of the manuscript. M.T.U., S.M., R.H., T.M., R.T., and A.K. gave critical advice throughout the experiments. T.A. contributed to human immunohistochemistry. A.S. contributed to primary mouse cortical neuron culture. T.U. conducted serial section transmission electron microscopy analysis. M.U. and U.K. contributed to study concept and edited the manuscript. All authors read and approved the final manuscript.

## Competing interests

The authors declare no competing interests.
