## [Peer Review File · Communications Biology]

Reviewers' comments:

Reviewer #1 (Remarks to the Author):

In this report, Asad-Utsugi et al., investigates the role of DNA double stranded breaks (DSBs) in Tau mediated AD pathology. The authors demonstrate a correlation between DSBs and phosphorylated Tau in post-mortem AD samples. Next, the authors used in-vitro induction of DSB using Etoposide and show increased Tau and p-Tau in the cytoplasm and nuclear periphery. Proximity ligation assays after DSB induction shows increased association of non-pTau with Tubulin indicating the association between Tau and microtubules in DNA repair. Finally, knockdown of Tau or addition of microtubule disassembly promoting compound exacerbated the effects of DSB induction. Overall, the study reveals important links between DSBs and Tau pathology that are relevant in AD pathogenesis. However, the authors should address a few concerns described below before the manuscript is considered for publication.

- 1) There is no quantification for the images shown in Figure 1.
- 2) The control and AD samples are not age-matched. More specifically, the average age of control samples is lower than AD by 15 years. Since DNA damage also increases with age, this is a major confounding factor for the quantification DNA damage in AD samples in this study.
- 3) There is no overlap between γ H2AX and neurons in the hippocampus (where most of the γ H2AX signal is present) of post-mortem AD samples. However, the rest of the study focuses on neurons. The authors attributes the lack of co-localization to early neuronal death. But previous studies that are cited here demonstrate the presence of γ H2AX within neurons in AD post-mortem samples (Thadathil et al. 2020, Shanbhag et al. 2019). Therefore, the authors should clearly discuss this before transitioning to the primary neuronal culture studies.
- 4) In Fig 2, the authors show high levels nuclear Tau after ETP treatment by western blot. But the ICC images in figure 3 and 4 show Tau accumulation predominately in the cytoplasm in response to ETP. 1) How do the authors reconcile this difference. 2) Can the authors quantify the cytoplasmic vs nuclear Tau levels based on ICC images as well.
- 5) In Figure 4C, why is the ETP concentration different (50uM)? Does Tau oligomerize at the 50uM ETP concentration that the authors use throughout the study?
- 6) The authors use LaminB1 to normalize the nuclear fraction in western blot experiments. However, ETP induced apoptosis can affect Lamin B1 levels and therefore Lamin B1 will not serve as an ideal loading control.
- 7) The authors suggest that while non-p-Tau aids DSB repair by associating with tubulin, p-Tau does not associate with Tubulin. Can the authors perform PLA assay similar to Fig 5 but using AT8 antibody to directly test this.
- 8) Finally, it's unclear from the model what triggers the switch between normal processing of DSBs via non-p-tau and the pathological processing mediated by p-Tau. Do the authors speculate that it's the level of DSBs that triggers the pathological pathway? This should be made clear in model figure legend.

Minor:

In the abstract the authors say deletion of endogenous Tau. Please clarify that it's a knockdown, not a deletion.

Reviewer #2 (Remarks to the Author):

In this manuscript, the authors study the DNA double strand breaks (DSB) and tau correlation in Alzheimer's disease using post-mortem AD and non-AD brains and in vitro model system. They show that DSBs increase in AD brains as determined by increase γ -H2A.X signal. Using cell culture studies, they further showed that DSB induction with etoposide causes accumulation of tau around nucleus which are then converted into phosphorylated tau. There are a few points that require revision in this study, which are itemized below. The results are of interest and they confirm previous reports of a connection between DNA lesions, tau and neurodegenerative processes. In Figure 1, the authors investigate colocalization of DSBs with MAP2 positive cells in postmortem

brain tissue; it would be interesting to show whether DSBs are present in other cells type such as glial/vascular cells in postmortem tissue. Shanbhag et al, and Thadathil et al for instance, have shown that DSBs increase in both neurons and astrocytes of frontal cortex and hippocampus of AD brains. A similar comparison in the temporal cortex and white matter is required.

How different fields were selected for H2A.X intensity?

Previous studies already reported that activation of DNA damage response causes the phosphorylation of tau and promotes neurodegeneration (Kanae Iijima-Ando et al; Kathleen M. Farmer, et al). It is conceivable that once DSBs occur, they recruit several DNA repair sensor and effector proteins which in turn activate signaling pathways to influence AD-related neuropathology, but the mechanisms how DSB accumulation trigger tau phosphorylation are largely unexplored. Under normal condition p53 associates with microtubules. It is not clear whether DSBs interfere with this p53-tau interaction during pathological condition. Does this effect on p53 activity and stability?

Does non-p tau interact with any perinuclear membrane proteins.

On page 19, author stated that endogenous mouse tau expression levels effectively reduced about ~20% of tau protein in comparison to control shRNA but figure 6 b shows ~ 60-7-% reduction in tau levels. In Fig. 6D it is not clear whether control cells are also treated with shRNA. It would be good to also show total H2Ax levels as loading control.

There is inconsistency in figure legend 6 d. please correct it.

Neurons in use are not fully matured neurons (7 days instead of the usual 12-14 days) but still in development. The choice should be explained in the text or the experiment should be repeated with matured primary neurons.

DNA double strand breaks result in small intranuclear foci of colocalized γ H2AX rather than in diffuse pan-nuclear staining as shown in this manuscript.

In fig 6, cells knockdown with tau shRNA followed by etoposide treatment for 30 min showed increased γ H2Ax (2-fold higher). However, 24 h after of 30 min of etoposide treatment causes ~60 % reduction in γ -H2A.X levels. It is well known that robust induction of γ H2A.X in cells treated with etoposide were observed by several prior studies and this induction get reduce after certain time. This reviewer had a hard to understand the authors conclusion of this experiment that the interaction between tau and tubulin contributes to DSB repair in the early phase, but the conversion from non-p-tau to p-tau by affecting tubulin polymerization in the chronic stage may affect the DNA repair system.

Reviewer #3 (Remarks to the Author):

The manuscript of Asada-Utsugi and colleagues is of great interest to the field of tauopathies and neurodegeneration in general. While a possible connection between DNA lesions and neurodegenerative processes is likely, formal proof of it is still missing. Using both human specimens and rodent primary neuronal cultures, the authors try to uncover a still missing mechanism behind the potential role of the protein tau in DNA repair processes. A possible role of microtubule disorganization in the repair of DNA double strand breaks is novel and of great interest if fully proven. However, although the study is of great interest to the field, it is lacking key and crucial experiments to really back up the claims. The manuscript would also benefit from a more detailed introduction and a more comprehensive description of the state of the art on the connection between tau and DNA repair.

Specific Comments:

1. Several articles have already been published on the possible role of tau in the repair of DNA lesions and not all the key articles have been cited in the text. The authors may want to add a few crucial references, such as PMID: 30531974 where they, for the first time, studied the connection between tau and the drug etoposide. A recent review has also been published on the topic PMID: 33297375; it may help the authors to pick some key citations that could be added to the text.
2. The field of DNA repair in neurons has also seen several great papers being published lately. Some of them have given a new dimension to the problem of DNA lesions in physiological and pathological brain aging, such as: PMID: 32424276, PMID: 31101070 or PMID: 29217584. The authors may consider adding some of them.
3. One of my major concern about the idea behind the study is that neurons do not replicate and

therefore the role of the microtubules in DNA repair may be different from what has already been published. The authors may consider adding some details about it in the introduction.

4. DIV7 neurons are considered immature primary cells. The authors should justify way they did not use mature primary neurons (DIV12-14) or consider replicating some of the experiments with mature cultures.

5. Give more context to chromosome movements in neurons. Perform ad hoc experiments to prove it such as in the cited publications.

6. Figure 1: quantifications could improve the data. The sample size is very limited.

7. Figure 1: would be good to analyze in all panels the same brain regions.

8. Table 1: control 4, cause of death Breast Cancer. Since it is a very rare condition in males, please check whether by any chance this is a mistake (it does not have to be thou).

9. Figure 2: since these are not cycling cells, it would be good to prove by COMET assay that DSBs are indeed formed.

10. Figure 2: N could be improved.

11. Figure 2: detergent insolubility assays could define whether pathological and hyperphosphorylation of tau also causes, or is linked to, aggregation of the protein. Please describe the phosphorylation status of tau in the nucleus in the context of the literature (PMID: 30531974).

12. Could the authors better explain the meaning of "top-tau" at page 16?

13. Figure 6: at least another shRNA targeting tau is required to avoid misinterpretation of data due to offside effects.

14. More mechanistic experiments are required to really prove that tau regulates DSBs through microtubule dynamics. Right now the manuscript is descriptive and with no clear mechanism. Experiments with tau mutants that do not bind tubulin could help in the understanding of the mechanism. Tau mutants that do not go to the nucleus could also be studied.

Points of revisions

	Before	After
Abstract	(P5, L7~) In vitro studies using primary cultured neurons showed that non-p-tau accumulated perinuclearly together with the tubulin after DSB induction with etoposide, followed by the conversion to p-tau. Moreover, the deletion of endogenous tau exacerbated DSB in neurons, suggesting the protective role of tau on DNA repair.	(P5, L7~) In vitro studies using primary cultured neurons showed that non-p-tau accumulated perinuclearly together with the tubulin after DSB induction with etoposide, followed by the accumulation of phosphorylated tau . Moreover, the knockdown of endogenous tau exacerbated DSB in neurons, suggesting the protective role of tau on DNA repair.
Average age of control and AD brain	control 70.8 yo AD 85.8 yo (Tabel 1)	control 78.2 yo AD 81.7 yo (Tabel 1)
Gender of control subject No.4	control 4, M, Breast Cancer (Tabel 1)	control 4, F, Breast Cancer (Tabel 1)
Quantification of γ H2Ax in autopsy brains	Immunohistochemistry images only (Fig.1)	We've added the quantification of γ H2Ax in cortex, hippocampus and white matter (Fig.1 b, g, h)
Immunofluorescence image and quantification data by cell type		We've added the regional difference of cell-specific vulnerability to DSB, and conducted the double immunofluorescent study, using antibodies against γ H2Ax vs. anti-NeuN, Iba1, GFAP, olig2,

		and ZO-1 for neurons, microglia, oligodendrocytes, and endothelial cells. Suppl.Fig.1
Loding control of WB in nuclear facion	LaminB (Fig.2 a)	The loading control of WB for nuclear fraction was changed to Histione H3, and the quantitative graph was rewritten. But, there was no significant differnce any data (Fig.2 a)
PLA assay data of non-p-tau or p-tau and nuclear membrane protein		We performed PLA assay for Tau1-H3K9me3, Tau1-LaminB, p-Tau(AT100)- H3K9me3. (Supp.Fig.4)
Quantification data of Tau1 density in DAPI area for 6hrs with etiposide treatment.	Fig3	We' ve added the qauntification data of Tau1 density in DAPI area for 6hrs with etoposide treatmen(Fig.3 d).
Quantification data of AT8 density in DAPI area for 24hrs with etoposide.	Fig4	We've added the quantification data of AT8 density in DAPI area for 24hrs with etoposide (Fig.4 b)
Etoposide concentration of oligomeric tau staining.	5 μ M etoposide-24hrs (Fig.4 c)	50 μ M etoposide-24hrs Fig.4 d
Loding control of WB in nuclear facion.	LaminB (Fig.2 a)	Histone H3 (Fig.2 a)
Hypothetical model of neuronal death and DNA damage repair from the perspective of the relationship between	Fig.8 and legend	I've changed the hypothetical model to show that excessive DSB levels lean towards pathological (Fig. 8).

tau and microtubule polymerization(fig.8)		
Knowkdown efficiency	(P15, L9~) Endogenous mouse tau expression levels effectively reduced about ~20% of tau protein in comparison to control shRNA (Figs.6 a, b, c).	(P13, L9~) Endogenous mouse tau expression levels effectively reduced about ~60% of tau protein compared to control shRNA (Figs.6 a, b, c).
Fig. 6 legend	(d) Left panel: Western blots of control and mouse tau shRNA knockdown with 50 μ M etoposide for 30 mins. Right panel: Confocal microscopy images of mouse tau (green), γ H2Ax (red), and DAPI (blue) with 50 μ M etoposide for 30 min. Scale bar = 100 μ m.	(d) Western blots of control and mouse tau shRNA knockdown with 50 μ M etoposide treatment for 30 mins.
Matture neuron culture (DIV 14)		This result was newly provided in Suppl.Fig.3 b, and a description regarding this was added in the revised manuscript.
Addition of references		PMID: 30531974; newly cited in the result section for Figure 2 as ref 48. PMID: 33297375; newly cited in the introduction section as ref 32. PMID: 32424276; newly cited in the introduction section as ref 6. PMID: 31101070; newly cited in

		the introduction section as ref 7. PMID: 29217584; newly cited in the introduction section as ref 5.
COMET assay		We've added COMET assay data for 0,0.5, 3, 6 and 24 hrs with etoposide treatment (Fig.2 c, d).
Addition of other shRNA clones		Similar results were obtained for other shRNA clones (Supplemental Fig. S7 c).
Change loading control of WB on mouse tau knockdown.	γ H2Ax / β actin (Suppl.Fig.4 b)	The loading control was changed to H2Ax and quantified again(Suppl.Fig.7 a, b)
Graphs		We converted all bar graphs to box-and-whisker and dot -plot format .

First of all, we appreciate the time and effort reviewers have spent reviewing this manuscript. The reviewers' comments are very thoughtful and constructive. We do honestly agree with them. We hope the responses below address the reviewers' comments satisfactorily.

Responses to Reviewer #1

We appreciate many important and rational suggestions of this reviewer, which may improve our manuscript. We are also grateful to know the encouraging comment of reviewer #1 "Overall, the study reveals important links between DSBs and Tau pathology that are relevant in AD pathogenesis."

1) There is no quantification for the images shown in Figure 1.

>>We thank this reviewer for the comment, which helps strengthen our data. We performed the quantification analysis of our immunohistochemical data, which successfully demonstrated the significant increase in the region-specific increase in the neuronal DSB. The results of the hippocampus, entorhinal cortex, and white matter were replaced to figures as Figure 1b, 1d, and 1h, respectively, together with the quantification data. The original and revised figures have been presented as follows.

Original figure:

Fig. 1 Accumulation of DSB marker in AD brains

(a) Representative images of immunohistochemistry using DAB stain for A β and γ H2Ax (phosphorylated histone variant H2Ax at residue Ser139, DSB marker) in the hippocampus. Scale bars=500 μ m for A β and 100 μ m for γ H2Ax, respectively. (b, c) Representative images of double immunohistochemistry using anti-MAP2 (neuronal marker, red) and anti- γ H2Ax (black) antibodies in the temporal lobe (TL) and hippocampus (Hip). Scale bar=50 μ m (d, e) Representative images of double immunohistochemistry for phospho-tau (AT8, pS202/T205, red) and γ H2Ax (black) in the cortex (Cx) and the white matter (Wm). Scale bar=50 μ m. DSB, DNA double-strand break; AD, Alzheimer's disease.

Revised figure:

Fig. 1 Accumulation of DSB marker in AD brains

(a) Representative images of immunohistochemistry using DAB stain for Aβ in the temporal lobe cortex (Cx). Representative images of double immunohistochemistry for anti-MAP2 (neuronal marker, red) and anti-γH2Ax (phosphorylated histone variant H2Ax at residue Ser139, DSB marker, black) in the hippocampus (Hip). Scale bars=200 μm for Aβ, 100 μm for MAP2 and γH2Ax, 50 μm for magnified images, respectively. (b) Quantitation of γH2Ax immunoreactivity in the hippocampus. ** $p = 0.0027$, The p -value was obtained by Student's t-test. (c, d) Representative images of double immunohistochemistry using anti-MAP2 (red) and anti-γH2Ax (black) antibodies in the entorhinal cortex (ECx). Scale bar=50 μm (e, f) Representative images of double immunohistochemistry for anti-MAP2 (red) and anti-γH2Ax (black) antibodies in the white matter (Wm). Scale bar=50 μm. (g) Quantitation of γH2Ax immunoreactivity in MAP2 positive neuron in the cortex and entorhinal cortex. * $p = 0.0177$, The p -value was obtained by Student's t-test. (h) Quantitation of γH2Ax immunoreactivity in the white matter. $p = 0.4619$, The p -value was obtained by Student's t-test. (i, j) Representative images of double immunohistochemistry for anti-AT8 (phosphorylated at S202/T205, red) and anti-γH2Ax (black) antibodies in the Cortex. Scale bar=50 μm. (k) Representative immunofluorescent images using antibodies against p-tau (AT8, green), γH2Ax (red), and DAPI (Blue) in the hippocampus. White squares are magnified images. Scale bar=100 μm. DSB, DNA double-strand break; AD, Alzheimer's disease.

2) The control and AD samples are not age-matched. More specifically, the average age of control samples is lower than AD by 15 years. Since DNA damage also increases with age, this is a major confounding factor for the quantification DNA damage in AD samples in this study.

>>We agreed with the comment pointing to the age mismatch between the AD and control subjects. Although preparation of non-neurological aged brain was a high hurdle, we additionally analyzed five other brains from three aged subjects with non-neurological diseases and those with juvenile AD. As a result, the mean age was 78.2 and 81.7 of the control and AD, respectively. This is summarized in Table1, as follows. For reference, the brain sections shown in Figure 1a were obtained from the subjects with 94 yo control and 93 yo AD.

Original table:

Table 1 Characteristics of human brain samples

	Case	Age at death(years)	Sex	Clinical diagnosis	NFT stage	CERAD
Control	1	74	F	Suffocation	0	B
	2	68	M	Rheumatoid arthritis	0	0
	3	70	F	Cerebral hemorrhage	0	0
	4	68	M	Breast cancer	0	0
	5	74	M	Lung cancer	III	B
AD	1	89	F	Alzheimer disease	V	C
	2	93	F	Alzheimer disease	V	C
	3	85	M	Alzheimer disease	VI	C
	4	77	F	Alzheimer disease	V	C
	5	85	M	Alzheimer disease	IV	C

Table 1. Characteristics of human brain samples

Clinical and histopathological information of the human brain samples used in this study. We analyzed five brains from patients with neuropathology-confirmed AD and five brains from non-neurodegenerative disease control subjects. NFT, neurofibrillary tangle; CERAD, Consortium to Establish a Registry for Alzheimer's Disease.

Revised table:

Table 1 Characteristics of human brain samples

	Case	Age at death(years)	Sex	Clinical diagnosis	NFT stage	CERAD
Control	1	74	F	Suffocation	0	B
	2	68	M	Rheumatoid arthritis	0	0
	3	70	F	Cerebral hemorrhage	0	0
	4	68	F	Breast cancer	0	0
	5	74	M	Lung cancer	III	B
	6	94	F	Disseminated intravascular coagulation	II	0
	7	92	M	Prostate cancer	III	0
	8	86	M	Intracranial hemorrhage	III	0
AD	1	89	F	Alzheimer disease	V	C
	2	93	F	Alzheimer disease	V	C
	3	85	M	Alzheimer disease	VI	C
	4	77	F	Alzheimer disease	V	C
	5	85	M	Alzheimer disease	IV	C
	6	76	F	Alzheimer disease	VI	C
	7	67	F	Alzheimer disease	VI	C

*Table legends were not changed.

3) There is no overlap between γ H2AX and neurons in the hippocampus (where most of the γ H2AX signal is present) of post-mortem AD samples. However, the rest of the study focuses on neurons. The authors attributes the lack of co-localization to early neuronal death. But previous studies that are cited here demonstrate the presence of γ H2AX within neurons in AD post-mortem samples (Thadathil et al. 2020, Shanbhag et al. 2019). Therefore, the authors should clearly discuss this before transitioning to the primary neuronal culture studies.

>>We agree with the comment, which is also a matter of discussion ourselves. According to the valuable comment, we focused on the regional difference of cell-specific vulnerability to DSB, and conducted the double immunofluorescent study, using antibodies against γ H2Ax vs. anti-NeuN, Iba1, GFAP, olig2, and ZO-1 for neurons, microglia, oligodendrocytes, and endothelial cells. The results showed that neuronal DSB was detected in temporal and entorhinal cortices, whereas the γ H2Ax-positive cells were non-neuronal in the hippocampus. Data were newly provided as Figures 1c, d, and d, and supplementary figure 1 (please see below). We believe that using primary cortical neurons for *in vitro* analyses will be rational to the human AD pathology,

New figure

Suppl.Fig.1

Suppl. Fig. 1 Cell-type predominancy of DSB in the cortex , hippocampus and white matter of AD brains.

Immunofluorescence images of γ H2Ax (red) with (a) NeuN (green) of neuronal marker , (b) GFAP (green) of astrocyte marker, (c) Iba1 (green) of microglia marker, (d) Olig2 (green) of oligodendrocyte marker and (i) ZO1 (green) of endothelial cell marker. White boxes are zoomed images of dash boxes. Scale bar = 50 μ m. (e) Pearson's correlation coefficient values showing predominant overlay between NeuN and γ H2Ax in the cortex (n=3, **p=0.001, ***p=0.0006), (f) between; γ H2Ax and GFAP in hippocampus (n=3, Cx vs. Hip; ***p=0.0003, Hip vs. Wm; ***p < 0.0001), (g) between; γ H2Ax and Iba1 (n=3, *p=0.0241, **p=0.0091), and (h) between; γ H2Ax and Olig2 in the white matter (n=3, Cx vs. Wm; ***p=0.0003, Hip vs. Wm; ***p = 0.0001). n: number of images examined; Statistical significance was determined by two-way ANOVA, followed by a Tukey test. Error bars represent SD.

4) In Fig 2, the authors show high levels nuclear Tau after ETP treatment by western blot. But the ICC images in figure 3 and 4 show Tau accumulation predominately in the cytoplasm in response to ETP. 1) How do the authors reconcile this difference. 2) Can the authors quantify the cytoplasmic vs nuclear Tau levels based on ICC images as well.

>1) We agree with the reviewer's comment. As the reviewer pointed out, cytosolic tau also seems increased, as high as nuclear tau. Since figures 3 and 4 suggest perinuclear accumulation of tau after ETP treatment, we

performed a PLA assay to see the association between nuclear membrane and non or p-tau. As a result, PLA signals were significantly increased by UV and ETP, indicating the enhanced interaction between the two by DSB. The result was newly provided as Suppl.Fig.4, as follows.

New figure

Suppl. Fig. 4 PLA assay for tau and nuclear membrane protein under DSB induction

(a) Left panels; PLA signals (red) for non-p-tau (Tau1) and heterochromatin marker (H3K9me3) with UV exposure for 10 mins in primary mouse cortical neuron cultures (DIV 7). Right panels; Phase-contrast images. Scale bar =10 μ m. **(b)** Quantification of PLA signals for colocalized Tau1-H3K9me3 with DAPI ; n=3, UV ; n=5, * p =0.0364, n: number of images examined; Student's T-test, error bars represent SD. **(c)** Upper panels; PLA signals (red) for Tau1 and LaminB at 0, 0.5, 3, 6 and 24 hrs after 50 μ M etoposide treatment into primary mouse cortical neuron cultures (DIV 7). Scale bar =30 μ m. Bottom panels; Magnified images of the white square box. **(d)** The number of PLA foci in primary mouse cortical neurons exposed to 50 μ M etoposide for 0 hr (n=3), 0.5 hrs (n=3), 3 hrs (n=3), 6 hrs (n=3) and 24 hrs (n=4). n: number of images examined; The p -value was determined by two-way ANOVA, followed by a Tukey test. Error bars represent SD. 0 hr vs 6 hrs., ** p =0.0014; 0.5 hrs vs 6 hrs., *** p =0.0006; 3hr vs 6 hrs., * p =0.0473; 6 hrs vs 24hrs., * p =0.0120. **(e)** PLA assay shows the interation between PHF tau (AT100) and H3K9me3 with UV exposure for 10mins in primary mouse cortical neuron cultures (DIV7). **(f)** Quantification of PLA signals for colocalized p-tau(AT100)-H3K9me3 with DAPI. * p = 0.032, n= 4.

>>2) As the reviewer pointed out, we quantified the cytoplasmic versus nuclear intensities of Tau1 and AT8 in etoposide-treated primary neuron cultured cells. The results are shown in the figure below.

Figure only for review
Left: Quantification show Tau1 levels of cytoplasmic vs nucleus for 0hr and 6hrs with 50 μ M etoposide treatment. Fig. 2 and 3 show that Tau1 levels of the nuclear fraction was highest in the 6-hour treatments. Blue columns: Cytoplasm (0hr; n=10, 6hrs; n=14), Red columns: Nucleus (0hr; n=10, 6hrs; n=14). **Right:** Quantification show AT8 intensity of cytoplasmic vs nucleus for 0hr and 24hrs with 50 μ M etoposide treatment. Fig. 2 and 3 show that AT8 levels of the nuclear fraction was highest in the 24hour treatment. Blue columns: Cytoplasm (0hr; n=8, 6hrs; n=11), Red columns: Nucleus (0hr; n=8, 6hrs; n=11). Statistical significance was determined by two-way ANOVA followed by a Tukey test. n: number of cells examined. Error bars represent SD; * p <0.05, ** p <0.001, *** p <0.0001.

Etoposide treatment increased tau intensity in both cytoplasm and nucleus. We hypothesized that the tau accumulation around the nuclear membrane shown in Fig.3 and Fig.4 might be important for DSB damage and repair. We attempted to address this point by quantifying the nuclear tau density of the primary neurons treated with 50uM ETO for 6 and 24 hours by counting Tau1- or AT8-positive neurons on DAPI staining. These quantification analyses were newly provided in Figure 3d and Figure 4b, as follows.

Original figure:

Revised figure:

Fig. 3 Increase in non-p-tau around the nuclear membrane of neurons by DSB induction

(a) Top panels: Representative XY, YZ, and XZ projections of immunofluorescent images using antibodies against non-p-tau (Tau-1, green), γ H2Ax (red), and DAPI (Blue). Mice primary neurons were treated with 50 μ M etoposide or PBS (0 μ M ETO) for 6 hrs (50 μ M ETO, 6 hrs). White boxes show magnified XZ projections of Tau-1 staining to clarify the distribution of non-p-tau. The bottom panels show 3D reconstruction images of immunostaining. Scale bar=10 μ m. **(b)** The population of γ H2Ax-positive neurons per total neurons labeled with DAPI in non-treatment (gray column, n=5) and 50 μ M etoposide-treatment for 6 hrs (red column, the number of images examined=7). n: number of images examined; ** $p = 0.0001$, The p -value was obtained by Student's t-test. **(c)** Quantification shows that the percentage of cells with Tau-1- γ H2Ax double-positive neurons were exclusively high in 50 μ M etoposide treatment (γ H2Ax+/Tau-1+). Statistical significance was determined by two-way ANOVA followed by a Tukey test. n: number of images examined. *** $p < 0.0001$. **(d) Quantification of the Tau-1 intensity in DAPI area. Gray column is non-treatment (n=4), red column (n=3) is 50 μ M etoposide treatment for 6hrs. n: number of images examined; ** $p = 0.0041$, The p -value was obtained by Student's t-test. **(e)** Immunogold electron micrographs in non-p-tau labeled with Tau-1 antibody from primary mouse cortical neurons with non-treatment (0 μ M etoposide) and 50 μ M etoposide treatment for 6 hrs Arrowheads indicate labeling the gold particles for Tau-1. N, Nucleus; n, nucleolus; M, Mitochondria. Scale bar=2 μ m.**

:

Fig.4

Revised figure:

Fig.4

Fig. 4 Accumulation of p-tau and toxic tau around the nuclear membrane of neurons by DSB induction
(a) Left panels: Representative XY, YZ and XZ projections of immunofluorescent staining using antibodies against p-tau (AT8, green) and γH2Ax (red) and DAPI (Blue) with non-treatment (0 μM etoposide) and 50 μM etoposide treatment for 24 hrs (50 μM Etoposide, 24 hrs). Right panels show magnified XZ projections of white solid dotted squares in the left panels. Scale bar=10 μm. **(b)** Quantification of the AT8 intensity in DAPI area. Gray column is non-treatment (n=3), red column (n=4) is 50 μM etoposide treatment for 24 hrs. n: number of images examined; * p = 0.0168, The p-value was obtained by Student's t-test. Error bars represent SD. **(c)** Immunogold electron micrographs from primary mouse cortical neurons with non-treatment (0 μM etoposide) and 50 μM etoposide treatment for 24 hrs (50 μM etoposide, 24 hrs.), showing AT8 positive p-tau around the nuclear membrane (AT8). Zoom panels show enlarged projections of dotted square and solid square. Arrowheads indicate gold particles for AT8 antibody. N, Nucleus; Blue dash line, nuclear membrane. Scale bar=2 μm. **(d)** Immunofluorescence using antibodies against oligomeric tau (T22, red) and γH2Ax (green) with DMSO treatment (left panel) and 50 μM etoposide treatment for 24 hrs (right panel). Scale bar=30 μm.

5) In Figure 4C, why is the ETP concentration different (5uM)? Does tau oligomerize at the 50uM ETP concentration that the authors use throughout the study?

>>We appreciate the reasonable comment. We newly provided the images of primary neurons exposed to 50uM ETO for 24 hrs in Figure 4d in the revised manuscript (see the figure above). In addition, to convince our results, we added the oligomerized tau in the neurons exposed to ETP of 0.5, 5, 25, and 50uM in the supplementary figure 5, as follows.

New figure:

Suppl. Fig. 5 DSB induction by etoposide increase oligomeric tau

Primary mouse cortical neurons (DIV 7) were exposed to 0, 5, 25, and 50 μ M etoposide for 24 hrs. Immunofluorescence using antibodies against oligomeric tau (T22, red) and γ H2Ax (green). Scale bar=50 μ m.

6) The authors use LaminB1 to normalize the nuclear fraction in western blot experiments. However, ETP induced apoptosis can affect Lamin B1 levels and therefore Lamin B1 will not serve as an ideal loading control.

>>We agree with the comment. We have changed the protein for the internal standard of nuclear fraction, from Lamin B to histone H3, as follows.

Original: figure

Revised figure:

7) The authors suggest that while non-p-Tau aids DSB repair by associating with tubulin, p-Tau does not associate with Tubulin. Can the authors perform PLA assay similar to Fig 5 but using AT8 antibody to directly test this.

>>We thank the reviewer for the valuable comment. According to the suggestion, we performed a PLA assay with an AT8 antibody. We successfully confirmed the marked loss of association between pTau and the αTubulin. We have added the new data in Suppl. Fig.6b, as follows.

New figure

Suppl.Fig.6

Suppl. Fig. 6 Different co-localization of non-p-tau and non-p-tau with α-tubulin under DSB induction with etoposide

(a) PLA signals (red) against non-p-tau (Tau1) and α-Tubulin. Primary mouse cortical neuron cultures (DIV 7) were treated for 0, 0.5, 3, 6, and 24 hrs with 50 µM etoposide. Bottom panels: PLA foci extraction images of PLA signals with DAPI. Scale bar=30µm. (b) PLA signals (red) against p-tau (AT8) and α-Tubulin, under the same same treatment, showing the lack of co-localization.

8) Finally, it's unclear from the model what triggers the switch between normal processing of DSBs via non-p-tau and the pathological processing mediated by p-Tau. Do the authors speculate that it's the level of DSBs that triggers the pathological pathway? This should be made clear in model figure legend.

>>We agree with the comment. From the results of several additional experiments, we have been more confident in the role of accumulation of DNA damage and microtubule depolymerization, which may result in DNA repair damage, p-tau accumulation, and neuronal apoptosis. We thank the reviewer for giving us this opportunity to examine this further. We made a minor revision of the model figure legends, as follows.

Original figure:

Revised figure:

Fig. 8 Hypothetical model of neuronal death and DNA damage repair from the perspective of the relationship between tau and microtubule polymerization

Upon DNA damage, non-p-tau (blue) binds to microtubules at the physiological level, which promotes microtubule (MT) polymerization and induces accumulation around the perinuclear membrane to support DNA damage repair (Right: normal condition).

When DNA damage reaches an excessive level, tau phosphorylation is enhanced and microtubule depolymerization occurs. Accumulation of p-tau (orange) around the perinuclear membrane suppresses the movement of damaged chromosomes or inhibits the recruitment of DNA damage repair factors into the nucleus by inhibiting nuclear cytoplasmic transport. Subsequently, insoluble p-tau accumulates in the cytoplasm and axons, leading to further microtubule depolymerization and neuron death (Left: pathological condition).

Minor:

In the abstract the authors say deletion of endogenous tau. Please clarify that it's a knockdown, not a deletion.

>>We corrected the term “deletion” to “knockdown” in the abstract of the revised manuscript.

Original manuscript: (P5, L9~) Moreover, the **deletion** of endogenous tau exacerbated DSB in neurons, suggesting the protective role of tau on DNA repair.

Revised manuscript: (P3, L9~) Moreover, the **knockdown** of endogenous tau exacerbated DSB in neurons, suggesting the protective role of tau on DNA repair.

Responses to Reviewer #2

We are grateful to learn this reviewer's high estimation of our manuscript. His/her positive comment, “The results are of interest, and they confirm previous reports of a connection between DNA lesions, tau and neurodegenerative processes,” is very encouraging.

1) In Figure 1, the authors investigate colocalization of DSBs with MAP2 positive cells in postmortem brain tissue; it would be interesting to show whether DSBs are present in other cells type such as glial/vascular cells in postmortem tissue. Shanbhag et al, and Thadathil et al for instance, have shown that DSBs increase in both neurons and astrocytes of frontal cortex and hippocampus of AD brains. A similar comparison in the temporal cortex and white matter is required.

>>We thank the valuable comments from this reviewer. According to the suggestion, we newly conducted the double immunofluorescent study to identify the type of cells which contain DSB, using antibodies against NeuN, GFAP, Iba1, Olig2, and ZO1, for neurons, astrocytes, microglia, oligodendrocytes, and endothelium. As a result, neuronal DSB was abundant in the cortex, whereas astrocytes were predominantly co-labeled with DSB marker in the hippocampus. As expected, DSB-positive cells were predominantly detected in oligodendrocytes in the white matter. Interestingly, microglia or vascular endothelium were almost DSB-negative. The micrographs and their quantification were newly provided in supplementary Fig.1 in the revised manuscript, as shown below.

New figure

Suppl.Fig.1

The additional experiments also found that γ H2Ax-positive cells were exclusively MAP2-positive in the temporal cortex, including the entorhinal cortex. The overlay rate was significantly high in AD patients. These results were newly provided in Figs.1 c,d,g in the revised manuscript, as follows.

Original figure:

Revised figure;

Fig.1

Fig. 1 Accumulation of DSB marker in AD brains

(a) Representative images of immunohistochemistry using DAB stain for A β in the temporal lobe cortex (Cx). Representative images of double immunohistochemistry for anti-MAP2 (neuronal marker, red) and anti- γ H2Ax (phosphorylated histone variant H2Ax at residue Ser139, DSB marker, black) in the hippocampus (Hip). Scale bars=200 μ m for A β , 100 μ m for MAP2 and γ H2Ax, 50 μ m for magnified images, respectively. (b) Quantitation of γ H2Ax immunoreactivity in the hippocampus. $p = 0.0027$, The p -value was obtained by Student's t-test. Error bars represent SD. (c, d) Representative images of double immunohistochemistry using anti-MAP2 (red) and anti- γ H2Ax (black) antibodies in the entorhinal cortex (ECx). Scale bar=50 μ m (e, f) Representative images of double immunohistochemistry for anti-MAP2 (red) and anti- γ H2Ax (black) antibodies in the white matter (Wm). Scale bar=50 μ m. (g) Quantitation of γ H2Ax immunoreactivity in MAP2 positive neuron in the cortex and entorhinal cortex. $p = 0.0177$, The p -value was obtained by Student's t-test. Error bars represent SD. (h) Quantitation of γ H2Ax immunoreactivity in the white matter. $p = 0.4619$, The p -value was obtained by Student's t-test. Error bars represent SD. (i, j) Representative images of double immunohistochemistry for anti-AT8 (phosphorylated at S202/T205, red) and anti- γ H2Ax (black) antibodies in the Cortex. Scale bar=50 μ m. (k) Representative immunofluorescent images using antibodies against p-tau (AT8, green), γ H2Ax (red), and DAPI (Blue) in the hippocampus. White squares are magnified images. Scale bar=100 μ m. DSB, DNA double-strand break; AD, Alzheimer's disease.

2) How different fields were selected for H2A.X intensity?

>>We thank this reviewer for the comment. We randomly selected the fields for the counting conducted by other persons who were not informed of the origin of the photos.

3) It is conceivable that once DSBs occur, they recruit several DNA repair sensor and effector proteins which in turn activate signaling pathways to influence AD-related neuropathology, but the mechanisms how DSB accumulation trigger tau phosphorylation are largely unexplored. Under normal condition p53 associates with microtubules. It is not clear whether DSBs interfere with this p53-tau interaction during pathological condition. Does this effect on p53 activity and stability?

>>We thank the reviewer for the valuable comment. As the reviewer pointed out, how DSB triggers hyperphosphorylation of tau is not clarified in the present study. It is quite interesting to test the involvement of p53 in the process. In addition, involvement of other phosphorylation enzymes, such as GSK3 β , p38, cdk5 are also possible. The important question, "How tau proteins are phosphorylated after DSB induction" is an important theme of future study. We added this point in the discussion, as follows.

(P17, L10~) DSB may initially increase non-p-tau for DNA repair, but later result in the formation of cytotoxic tau species such as oligomeric tau or NFT through the activation of tau-phosphorylated enzymes such as CDK5, GSK3 β (glycogen synthase kinase 3 β) or p38 MAPK (mitogen-activated protein kinase). Thus, further investigation regarding the role of tau-phosphorylated enzymes in DSB induction in tauopathy will contribute to understanding AD.

4) Does non-p tau interact with any perinuclear membrane proteins.

>>We thank the reviewer for the critical comment. We newly performed PLA assay to investigate whether non-p-tau interacts with the nuclear membrane. First, the UV exposure augmented the PLA signals for tau1 and H3K9me3, a heterochromatin marker. Moreover, 50µM of etoposide (ETO) also enhanced the interaction between tau1 and LaminB, a nuclear membrane that is most prominent at 6 hrs after the ETO challenge. These results were newly provided in the revised manuscript in supplementary figures 4 (a,b for UV, c,d for ETO), as follows.

New figure

5) On page 19, author stated that endogenous mouse tau expression levels effectively reduced about ~20% of tau protein in comparison to control shRNA but figure 6 b shows ~ 60-7-% reduction in tau levels.

>>We made a typing error and corrected the reduction by about 60% in the revised manuscript. We thank this reviewer for avoiding the confusion of readers.

Original manuscript: (P19, L6~) Endogenous mouse tau expression levels effectively reduced about ~20% of tau protein in comparison to control shRNA (Figs.6 a, b, c).

Revised manuscript: (P13, L9~) Endogenous mouse tau expression levels effectively reduced about ~60% of tau protein compared to control shRNA (Figs.6 a, b, c).

6) In Fig. 6D it is not clear whether control cells are also treated with shRNA.

>>We thank the reviewer for the comment. Control cells were treated by viral vector control(pLKO.1). We apologize for causing confusion.

7) It would be good to also show total H2Ax levels as loading control.

>>As suggested by the reviewer, we used H2Ax as a loading control and expressed it as γ H2Ax / H2Ax to obtain quantitative data. The new data was provided in Fig.6 d,e in the revised manuscript, as follows.

Revised figure

8) There is inconsistency in figure legend 6 d. please correct it.

>>We appreciate the notice of inconsistency. We deleted the confusing and redundant lines in red in the original manuscript, as follows.

Original manuscript: (d) **Left panel:** Western blots of control and mouse tau shRNA knockdown with 50 μ M etoposide for 30 mins. **Right panel:** Confocal microscopy images of mouse tau (green), γ H2Ax (red), and DAPI (blue) with 50 μ M etoposide for 30 min. Scale bar = 100 μ m.

Revised manuscript: (d) Western blots of control and mouse tau shRNA knockdown with 50 μ M etoposide treatment for 30 mins.

9) Neurons in use are not fully matured neurons (7 days instead of the usual 12-14 days) but still in development. The choice should be explained in the text or the experiment should be repeated with matured primary neurons.

>>We appreciate the valuable comment. As suggested by the reviewer, we performed a similar experiment, using the 14-day cultured neurons, and observed a dose-dependent increase of non-p-tau (Tau1 staining) in the nuclear fractions. We are thus confident our results obtained from primary neurons of 7 day-culture are compelling and applicable to more matured neurons. This result was newly provided in Suppl.Fig.3 b, and a description regarding this was added in the revised manuscript, as follows.

Revised manuscript

(P8, L9~) DSB induces tau accumulation in the nuclear fraction

Previous reports demonstrate that the DNA moves to the nuclear pores and interacts with the inner nuclear membrane proteins for DNA repair upon severe DSB. Dissociation of damaged DNAs from the unimpaired DNA prevents aberrant recombination^{21, 45, 46, 47}. Hence, we examined whether subcellular localization of tau is altered in neurons when DSB occurs using etoposide, a topoisomerase II inhibitor. Primary mouse cortical neurons (DIV 7) were treated using 50 μ M etoposide for 0, 0.5, 3, 6, and 24 hrs, and the cell lysates were separated into the cytoplasmic and nuclear fractions for Western blotting. The γ H2Ax levels were highest at 3 hrs after treatment and decreased subsequently (Figs. 2a, b). In addition, the comet assay to quantify DSB levels showed that the damage was highest after 3 hrs of etoposide exposure, which is consistent with the result of WB (Figs.2c, d). From the preliminary lethality evaluation, 24 hrs of exposure to 50 μ M etoposide reduced cell viability by about 60 % (Supplemental Fig. S2). Active Caspase3 was detected at 6 and 24 hrs after treatment in cytoplasm fraction (Fig. 2a). In the cytoplasm, p-tau (AT8) gradually elevated at 6, and 24 hrs after treatment, whereas non-p-tau (Tau-1; dephosphorylated at S195/S198/S199/S202) levels were the highest at 6 hrs and declined at 24 hrs after DSB (Figs. 2a, e). In the nuclear fraction, p-tau and non-p-tau showed a similar expression profile, which increased at 6 hrs after treatment (Figs. 2a, f). This result was similar to a previous study that etoposide treatment increased total tau in the nucleus⁴⁸. On the contrary, DSB induction

by ultraviolet (UV) exposure for 5- and 10-mins increased tau species, including total, non-phosphorylated, and phosphorylated tau in both nuclear and cytoplasmic fractions at 5 mins (Supplemental Fig. S3a). These results indicate that DSB induces nuclear accumulation of non-p-tau, which subsequently increases p-tau. UV exposure might accelerate tau accumulation due to the disproportionate effect. **Since primary mouse cortical neurons cultured for 14 days also showed similar results that non-p-tau and p-tau in the nuclear fraction increased in an etoposide concentration-dependent manner (Supplemental Fig. S3b). We decided to use primary neurons of DIV 7 for the following experiments.**

Suppl. Fig. 3 Increase of Tau in the nuclear fraction following DSB induction with UV exposure and etoposide treatment onto mature neurons.

(a) Primary mouse cortical neuron cultures (DIV 7) in 10 cm dish were exposed to UV light by the UV transilluminator (ATTO), separated into nuclear and cytoplasmic fractions, and were subjected to western blot analysis. H3K9me3, Heterochromatin marker; LaminB, nuclei marker; GAPDH, cytoplasm marker. **(b)** Primary mouse cortical neuron cultures (DIV 14) were treated with etoposide in a dose-dependent manner.

10) DNA double strand breaks result in small intranuclear foci of colocalized γH2AX rather than in diffuse pan-nuclear staining as shown in this manuscript.

>>We agree with the comment. We assume that 50μM EPO for 6 or 24 hrs may cause robust DSB, which may be responsible for the diffuse pan-nuclear staining of γH2AX. Indeed, γH2Ax staining under 30 min treatment with EPO showed punctuated foci. **Please refer to the figure only for review below.**

Figure only for review

Top panels:

Immunofluorescent images using antibodies against p-tau (AT8, green), γH2Ax (red), and DAPI (Blue). Primary mouse cortical neuron culture (DIV7) were no treatment.

Bottom panels : Primary mouse cortical neuron culture (DIV7) were treated for 30 mins with 50μM etoposide. Scale bar = 10μm.

11) *In fig 6, cells knockdown with tau shRNA followed by etoposide treatment for 30 min showed increased γH2Ax (2-fold higher). However, 24 h after of 30 min of etoposide treatment causes ~60 % reduction in γ-H2A.X levels. It is well known that robust induction of γH2A.X in cells treated with etoposide were observed by several prior studies and this induction get reduce after certain time.*

>>We appreciate this thoughtful and valuable comment. We consider that the difference in temporal recovery after DSB depends on the type of cells. The natural reduction of γH2Ax after transient DSB may occur in the dividing cells due to the potent DNA repair through homologous recombination. On the other hand, the non-dividing primary neurons may have different resilience to DSB since we observed considerable death of neurons even after the short-term exposure to ETO and bathed in the normal nutrient medium for a further 24hr.

12) *This reviewer had a hard to understand the authors conclusion of this experiment that the interaction between tau and tubulin contributes to DSB repair in the early phase, but the conversion from non-p-tau to p-tau by affecting tubulin polymerization in the chronic stage may affect the DNA repair system.*

>>We appreciate this critical comment. As the reviewer pointed out, our results are mainly descriptive, and there is a gap between tau phosphorylation and microtubule disassembly as well as derangement of DNA repair system. However, one of our main findings is that pharmacological depolymerization of microtubule leads to 1) enhanced EPO toxicity such as DNA damage and apoptosis and 2) hyperphosphorylation of tau and tau aggregation resembling AD pathology (Figure 7). Since the promotion of microtubule assembly is an important physiological role of tau protein and phosphorylation of tau compromises this activity, we speculate that accumulation of p-tau might play an essential role in DNA damage observed in AD pathology.

Responses to Reviewer #3

We are grateful for the positive comments of reviewer #3, especially his/her comment, “The manuscript of Asada-Utsugi and colleagues is of great interest to the field of tauopathies and neurodegeneration in general”, is quite encouraging.

1) *Several articles have already been published on the possible role of tau in the repair of DNA lesions and not all the key articles have been cited in the text. The authors may want to add a few crucial references, such as PMID: 30531974 where they, for the first time, studied the connection between tau and the drug etoposide. A recent review has also been published on the topic PMID: 33297375; it may help the authors to pick some key citations that could be added to the text.*

>>We appreciate the reviewer’s instruction of the previous articles we had missed. We have added the following two works to the revised manuscript.

PMID: 30531974; newly cited in the result section for Figure 2 as ref 48.

PMID: 33297375; newly cited in the introduction section as ref 32.

2) *The field of DNA repair in neurons has also seen several great papers being published lately. Some of them have given a new dimension to the problem of DNA lesions in physiological and pathological brain aging, such as: PMID: 32424276, PMID: 31101070 or PMID: 29217584. The authors may consider adding some of them.*

>>We also thank the reviewer for showing us the great achievements we have missed in the original manuscript. These three papers have additionally been cited in the revised manuscript.

PMID: 32424276; newly cited in the introduction section as ref 6.

PMID: 31101070; newly cited in the introduction section as ref 7.

PMID: 29217584; newly cited in the introduction section as ref 5.

3) *One of my major concern about the idea behind the study is that neurons do not replicate and therefore the role of the microtubules in DNA repair may be different from what has already been published. The authors may consider adding some details about it in the introduction.*

>>This comment is a really reasonable and essential issue, and we appreciate it. We agree that we should clarify our working hypothesis to avoid confusion of the readers, we have modified the line in the revised manuscript as follows.

Original manuscript: (P7, L1~) Thus, DNA damage repair and chromosome movement are closely related, while cytoskeleton proteins are responsible for this system. Although

evidence indicates that DNA damage is implicated in the pathogenesis of neurodegenerative diseases^{1, 22, 23, 24, 25, 26}, **the exact role of DNA repair and the chromosome movement in neurons is unclear.**

Revised manuscript: (P5, L2~) Thus, DNA damage repair and chromosome movement are closely related, while cytoskeleton proteins are responsible for this system. Although evidence indicates that DNA damage is implicated in the pathogenesis of neurodegenerative diseases^{1, 22, 23, 24, 25, 26}, **the exact role of mobility-dependent DNA repair and the microtubule in neurons is unclear.**

4) DIV7 neurons are considered immature primary cells. The authors should justify way they did not use mature primary neurons (DIV12-14) or consider replicating some of the experiments with mature cultures.

>>We thank the reviewer for raising the fundamental issue to interpret our results. As we already explained to reviewer #2 in comment 9), we had performed the experiment using primary mouse neurons at DIV14, and obtained similar results, which have been proved as Suppl.Fig.3 b in the revised manuscript. We appreciate you refer to the data.

Revised figure.

5) Give more context to chromosome movements in neurons. Perform ad hoc experiments to prove it such as in the cited publications.

>>We agree with the reviewer's comment. Visualizing the chromosome movement is an ideal strategy to show how the chromosome is involved in DNA repair. However, this is one big category and beyond our scope in the present manuscript, in which microtubule-tau assembly in the cytosol is focused. We thank the reviewer for giving us an important hint for our future research. In this 50 μ M etoposide treatment, the blots were thinner due to more cell death.

6) Figure 1: quantifications could improve the data. The sample size is very limited.

>>We thank the reviewer for the constructive comment. We did our best to increase the number of subjects, especially from ten to fifteen (please see the table below). We have newly performed the counting of γ H2Ax-positive cells in the brains from control and AD subjects, which have been newly provided in Fig. 1b in the revised manuscript. In addition, the quantitative results of γ H2Ax-MAP2 double-positive cells in the cortex and γ H2Ax-cells in the white matter have shown in Fig. 1g in the revised manuscript.

Original table

Table 1 Characteristics of human brain samples

	Case	Age at death(years)	Sex	Clinical diagnosis	NFT stage	CERAD
Control	1	74	F	Suffocation	0	B
	2	68	M	Rheumatoid arthritis	0	0
	3	70	F	Cerebral hemorrhage	0	0
	4	68	M	Breast cancer	0	0
	5	74	M	Lung cancer	III	B
AD	1	89	F	Alzheimer disease	V	C
	2	93	F	Alzheimer disease	V	C
	3	85	M	Alzheimer disease	VI	C
	4	77	F	Alzheimer disease	V	C
	5	85	M	Alzheimer disease	IV	C

Revised table

Table 1 Characteristics of human brain samples

	Case	Age at death (years)	Sex	Clinical diagnosis	NFT stage	CERAD
Control	1	74	F	Suffocation	0	B
	2	68	M	Rheumatoid arthritis	0	0
	3	70	F	Cerebral hemorrhage	0	0
	4	68	F	Breast cancer	0	0
	5	74	M	Lung cancer	III	B
	6	94	F	Disseminated intravascular coagulation	II	0
	7	92	M	Prostate cancer	III	0
	8	86	M	Intracranial hemorrhage	III	0
AD	1	89	F	Alzheimer disease	V	C
	2	93	F	Alzheimer disease	V	C
	3	85	M	Alzheimer disease	VI	C
	4	77	F	Alzheimer disease	V	C
	5	85	M	Alzheimer disease	IV	C
	6	76	F	Alzheimer disease	VI	C
	7	67	F	Alzheimer disease	VI	C

Original figure:

Fig.1

Fig. 1 Accumulation of DSB marker in AD brains

(a) Representative images of immunohistochemistry using DAB stain for A β and γ H2Ax (phosphorylated histone variant H2Ax at residue Ser139, DSB marker) in the hippocampus. Scale bars=500 μ m for A β and 100 μ m for γ H2Ax, respectively. (b, c) Representative images of double immunohistochemistry using anti-MAP2 (neuronal marker, red) and anti- γ H2Ax (black) antibodies in the temporal lobe (TL) and hippocampus (Hip). Scale bar=50 μ m (d, e) Representative images of double immunohistochemistry for phospho-tau (AT8, pS202/T205, red) and γ H2Ax (black) in the cortex (Cx) and the white matter (Wm). Scale bar=50 μ m. DSB, DNA double-strand break; AD, Alzheimer's disease.

Revised figure:

Fig. 1 Accumulation of DSB marker in AD brains

(a) Representative images of immunohistochemistry using DAB stain for A β in the temporal lobe cortex (Cx). Representative images of double immunohistochemistry for anti-MAP2 (neuronal marker, red) and anti- γ H2Ax (phosphorylated histone variant H2Ax at residue Ser139, DSB marker, black) in the hippocampus (Hip). Scale bars=200 μ m for A β , 100 μ m for MAP2 and γ H2Ax, 50 μ m for magnified images, respectively. **(b)** Quantitation of γ H2Ax immunoreactivity in the hippocampus. $^{**}p = 0.0027$, The p -value was obtained by Student's t-test. Error bars represent SD. **(c, d)** Representative images of double immunohistochemistry using anti-MAP2 (red) and anti- γ H2Ax (black) antibodies in the entorhinal cortex (ECx). Scale bar=50 μ m **(e, f)** Representative images of double immunohistochemistry for anti-MAP2 (red) and anti- γ H2Ax (black) antibodies in the white matter (Wm). Scale bar=50 μ m. **(g)** Quantitation of γ H2Ax immunoreactivity in MAP2 positive neuron in the cortex and entorhinal cortex. $^{*}p = 0.0177$, The p -value was obtained by Student's t-test. Error bars represent SD. **(h)** Quantitation of γ H2Ax immunoreactivity in the white matter. $p = 0.4619$, The p -value was obtained by Student's t-test. Error bars represent SD. **(i, j)** Representative images of double immunohistochemistry for anti-AT8 (phosphorylated at S202/T205, red) and anti- γ H2Ax (black) antibodies in the Cortex. Scale bar=50 μ m. **(k)** Representative immunofluorescent images using antibodies against p-tau (AT8, green), γ H2Ax (red), and DAPI (Blue) in the hippocampus. White squares are magnified images. Scale bar=100 μ m. DSB, DNA double-strand break; AD, Alzheimer's disease.

7) *Figure 1: would be good to analyze in all panels the same brain regions.*

>>We thank the reviewer for giving us helpful comments to improve our data. We retok the photos of the same regions and analyzed them for quantification. Please see the new Figure 1 above.

8) *Table 1: control 4, cause of death Breast Cancer. Since it is a very rare condition in males, please check whether by any chance this is a mistake (it does not have to be thou).*

>>We found we have misstyped as the reviewer pointed out. We corrected the sex of control #4 in the revised manuscript. Please refer to the tables above in comment 6).

9) *Figure 2: since these are not cycling cells, it would be good to prove by COMET assay that DSBs are indeed formed.*

>>As suggested by the reviewer, we have newly performed a COMET assay and successfully observed DSB in the neurons by etoposide. The new data have provided in Fig 2, c,d in the revised manuscript, as follows.

Original:

Fig.2

Fig. 2 Tau increase in the nuclear fraction by DSB induction

(a) Western blot analysis of tau in cytoplasmic and nuclear fractions from primary mouse cortical neuron lysates after 50 μM etoposide treatment. GAPDH and LaminB are used as markers for cytoplasmic and nuclear fractions, respectively. (b) Densitometric analysis of γH2Ax blots in (a). The p -value was obtained by one-way ANOVA followed by a Tukey test (Error bars represent SD, $n=4$). (c, d) The densitometric analysis of non-p-tau (Tau-1) in the cytoplasm (c, blue column) and the nucleus (d, red column), normalized by GAPDH and LaminB, respectively. The p -value was obtained by one-way ANOVA followed by a Tukey test (Error bars represent SD, $n=5$). (e, f) The densitometric analysis of p-tau (AT8) in the cytoplasm (e, blue column) and the nucleus (f, red column), normalized by GAPDH and LaminB, is shown. The p -value was obtained by one-way ANOVA followed by a Tukey test (Error bars represent SD, $n=5$).

Fig.2

Fig. 2 Tau increase in the nuclear fraction by DSB induction

(a) Western blot analysis of tau in cytoplasmic and nuclear fractions from primary mouse cortical neuron lysates after 50 μ M etoposide treatment for 0, 0.5, 3, 6, and 24 hrs. GAPDH and Histone H3 are used as markers for cytoplasmic and nuclear fractions, respectively. (b) Densitometric analysis of γ H2Ax blots in (a). The p -value was obtained by one-way ANOVA followed by a Tukey test ($P=0.0485$, $n=5$ independent experiments). (c) The levels of DNA damage are demonstrated by Alkaline comet images after 50 μ M etoposide treatment for 0, 0.5, 3, 6, and 24 hrs. (d) Mean normalized olive tail moment of comets is shown in primary mouse cortical neuron treated with 50 μ M etoposide treatment for 0, 0.5, 3, 6, and 24 hrs. The p -value was obtained by one-way ANOVA followed by a Tukey test. $*P < 0.05$, $**P < 0.01$, $***P < 0.001$, 0 hr $n = 67$, 0.5 hrs; $n=31$, 3 hrs; $n=52$, 6 hrs; $n=62$, 24 hrs; $n=42$. n ; the number of images examined. (e) The densitometric analysis of non-p-tau (Tau-1, white column) and p-tau (AT8, blue column) in the cytoplasm, normalized by GAPDH. $n=5$ independent experiments. (f) The densitometric analysis of Tau1 (white column) and AT8 (red column) in the nucleus, normalized by Histone H3. $n=5$ independent experiments.

10) Figure 2: N could be improved.

>>We completely agree with the reviewer's suggestion as for Fig.2e,f, and we redid our experiments. Although we confirmed a similar trend each time, the tau blot is quite fragile and makes us difficult to obtain statistical significance. Another reason could be that perinuclear tau is separated into nuclear or cytosolic fractions depending on the adherence with the nucleus membranes. Therefore, we focused on the quantitative immunofluorescent studies in Figure 3, and we appreciate that the reviewer could accept our strategy.

11) Figure 2: detergent insolubility assays could define whether pathological and hyperphosphorylation of tau also causes, or is linked to, aggregation of the protein. Please describe the phosphorylation status of tau in the nucleus in the context of the literature (PMID: 30531974).

>>We appreciate the important comment, and kind instruction of the crucial works to be cited. We have added the following lines in the Discussion section citing this reference, as follows.

(P16, L17~) Giorgio U. et al. also examined the phosphorylation state of nuclear tau in detail in proliferating pluripotent neuronal C17.2 and neuroblastoma SY5Y cells and showed an increase in pT181, pT212, and S404 48. DSB by etoposide decreased p-tau and increased non-p-tau, resulting in tau translocation into the nucleus. Conversely, we showed that DSB induction increases nuclear p-tau (AT8). The apparent discrepancy might be derived from the phosphorylation sites since we used the AT8 antibody. Otherwise, our results that DSB induction increases nuclear non-p-tau agree with theirs.

12) Could the authors better explain the meaning of “top-tau” at page 16?

>>As the reviewer pointed out, we have mistyped top-tau for p-tau. “to p-tau” could 41rroneously expressed as top-tau. We corrected it in the revised manuscript as follows, and thank the reviewer.

Original manuscript (P15, L14~): These results imply that the dynamic change of cytoplasmic non-p-tau may be involved in DNA repair through access to the nuclear membrane, whereas conversion **top**-tau is neurotoxic.

Revised manuscript (P12, L2~):

Moreover, UV exposure for 10mins also increased PLA foci for PHF-tau (AT100; pThr212/Ser214) and heterochomatin (Supplemental Figs. S4e, f).The induction of DSB with etoposide increased the toxic oligomeric tau stained with anti-T22 antibody in a dose-dependent manner, indicating that DSB-relevant perinuclear p-tau accumulation is linked to neurodegeneration in tauopathy (Fig.4d, Supplemental Fig. 5).

epair through access to the nuclear membrane, whereas conversion **p**-tau is neurotoxic.

13) Figure 6: at least another shRNA targeting tau is required to avoid misinterpretation of data due to offside effects.

>>We agree with the reviewer’s comment. As suggested, we performed shRNA experiments using the other two clones, which were provided in Suppl.Fig.7c in the revised manuscript.

Original figure:

Suppl. Fig. 4 The effect of mouse tau knockdown on DSB accumulation for long term

Primary mouse cortical neuron cultures (DIV 0) were transduced by mouse tau and control shRNA lentivirus particles, cultured for seven days. **(a)** Western bolts of primary neuron samples were treated with 50 μ M etoposide for 30 mins, followed by a culture for 24 hrs. with an etoposide-free medium. Casp3, total caspase 3; C-cas3, cleaved-caspase 3. **(b)** Quantitation of γ H2Ax/ β actin intensity for Western blot bands. Error bars represent SD, t-test, * $p=0.002$, $n=3$.

Revised figure

Suppl.Fig.7

Suppl. Fig. 7 The effect of mouse tau knockdown on DSB accumulation for long term

Primary mouse cortical neuron cultures (DIV 0) were transduced by mouse tau and control shRNA lentivirus particles, cultured for seven days. **(a)** Western blots of primary neuron samples were treated with 50 μ M etoposide for 30 mins, followed by a culture for 24 hrs with an etoposide-free medium. scr, control shRNA; sh1, mouse tau shRNA clone 1, Casp3, total caspase 3; C-cas3, cleaved-caspase 3. **(b)** Quantitation of γ H2Ax/ H2Ax intensity for Western blot bands, t-test, * $p=0.002$, $n=3$. **(c)** Western blots showing the DSB induction, caspase activation by etoposide in primary neuron samples pretreated with three different shRNA against tau. The shRNA-treated neurons were exposed to 50 μ M etoposide for 30 mins, subsequently replaced with an etoposide-free medium for another 24 hrs: sh2, mouse tau shRNA clone 2, sh3, mouse tau shRNA clone 3.

14) More mechanistic experiments are required to really prove that tau regulates DSBs through microtubule dynamics. Right now the manuscript is descriptive and with no clear mechanism. Experiments with tau mutants that do not bind tubulin could help in the understanding of the mechanism. Tau mutants that do not go to the nucleus could also be studied.

>>We thank the reviewer for the valuable comment. According to the suggestions, we performed cell transfection study to investigate the role of tau-microtubule interaction in DSB. HEK 293A cells were transfected with mutant tau, which is reported to lose microtubule binding affinity (human tau 1N4R; A123T). Unfortunately, we have found that DSB in the immortalized cells, observed at three hours after 10 μ M etoposide, are too fragile to obtain stable γ H2Ax expression among experiments. Moreover, the transient transduction of WT tau, tau A123T, together with EGFP-tubulin into HEK293A cells, showed comparable levels of γ H2Ax, as much as vector control (Results were shown in figures below only for review). Indeed, HEK cells display considerable DSB in the standard culture condition. Immortalized dividing feature of HEK cells might make us challenging to draw conclusive results. We next conducted primary neuron culture experiments, which were subjected to transient transfection with the tau 1N4R(A123T). Confocal microscope observation for γ H2Ax staining yielded a similar DSB appearance between WT and A123T tau (See the figures below only for review). We consider that endogenous tau in neurons was responsible for the results. The knock-in study using the primary neuronal cells is required, which we hope the reviewer understands, is an additional future challenge. Additionally, the time-lapse microscope observation of the primary neurons after DSB induction for microtubules, tau, and chromosomes will help us to clarify further the role of tau and microtubule assembly in the DNA repair. This is also our future change.

f)

a) PLA signals (red) for Myc (Myc-Tau 1N4R; WT) or (Myc-Tau 1N4R A123T; mutant) and β -Tubulin in HEK293A. Scale bar =50 μ m. **b)** Quantification of PLA signals for colocalized Myc- β Tubulin for number of DAPI ; WT=6, A123T ; n=4, * p =0.0229, n: number of cell; Student's T-test, error bars represent SD. **c)** Western blot analysis of HEK293A cells were transfected empty vector (pCMVMyc-N), Myc-Tau 1N4R (WT) and Myc-Tau 1N4R A123T with EGFP-Tubulin. **d)** Western blot analysis of HEK293A cells were transfected empty vector, WT and A123T with EGFP-Tubulin, treated for 3hrs with 10 μ M etoposide. **e)** Immunofluorescent images of EGFP (green), Myc (red), γ H2Ax (purple) and DAPI (blue) on HEK293A cells, transfected empty vector (pCMVMyc-N), EGFP-Tubulin, Myc (Myc-Tau 1N4; WT) or (Myc-Tau 1N4R A123T; mutant). Scale bar =50 μ m. **f)** Immunofluorescent images of primary mouse cortical neuron cultures (DIV 10), transfected Myc (Myc-Tau 1N4R; WT) or (Myc-Tau 1N4R A123T; mutant). Myc (red), γ H2Ax (purple) and DAPI (blue). Scale bar =50 μ m.